# A calibratable sensory neuron based on epitaxial VO$_2$ for spike-based neuromorphic multisensory system

Rui Yuan[1], Qingxi Duan[1], Pek Jun Tiw[1], Ge Li[2], Zhuojian Xiao[1], Zhaokun Jing[1], Ke Yang[1], Chang Liu[1], Chen Ge [2], Ru Huang [1,3,4✉] & Yuchao Yang [1,3,4,5✉]

Neuromorphic perception systems inspired by biology have tremendous potential in efficiently processing multi-sensory signals from the physical world, but a highly efficient hardware element capable of sensing and encoding multiple physical signals is still lacking. Here, we report a spike-based neuromorphic perception system consisting of calibratable artificial sensory neurons based on epitaxial VO$_2$, where the high crystalline quality of VO$_2$ leads to significantly improved cycle-to-cycle uniformity. A calibration resistor is introduced to optimize device-to-device consistency, and to adapt the VO$_2$ neuron to different sensors with varied resistance level, a scaling resistor is further incorporated, demonstrating cross-sensory neuromorphic perception component that can encode illuminance, temperature, pressure and curvature signals into spikes. These components are utilized to monitor the curvatures of fingers, thereby achieving hand gesture classification. This study addresses the fundamental cycle-to-cycle and device-to-device variation issues of sensory neurons, therefore promoting the construction of neuromorphic perception systems for e-skin and neurorobotics.

[1] Beijing Advanced Innovation Center for Integrated Circuits, School of Integrated Circuits, Peking University, Beijing 100871, China. [2] Beijing National Laboratory for Condensed Matter Physics, Institute of Physics, Chinese Academy of Sciences, Beijing 100190, China. [3] Center for Brain Inspired Chips, Institute for Artificial Intelligence, Peking University, Beijing 100871, China. [4] Center for Brain Inspired Intelligence, Chinese Institute for Brain Research (CIBR), Beijing, Beijing 102206, China. [5] Beijing Academy of Artificial Intelligence, Beijing 100084, China. ✉email: ruhuang@pku.edu.cn; yuchaoyang@pku.edu.cn

As the development of wearable electronics and internet of things (IoT), there is a dramatic upsurge in the type and number of sensory nodes[1], generating a great deal of sensory data that must be processed efficiently and in real time. In traditional architectures, the analog data collected by the sensors are first converted into digital signals via analog to digital converters (ADCs) and then stored in memory, before being forwarded to the computing units[2], hence causing high energy consumption and low efficiency, which is dramatically different from the highly efficient sensory processing of human. Human could sense the real world and outperform current digital systems in efficiency, robustness, flexibility, and fault tolerance[3]. The sensory system of human combines a variety of senses that work together and interact with the brain to allow people to explore and capture information[4–8]. In the human perception system, receptors receive physical stimuli from the outside world and convert physical information into electrical spikes, which are then delivered to the cerebral cortex of the brain for further processing[9]. This structure forms the basis of comprehensive perception, pre-processing, and coding capabilities of biological systems. To enable a biologically inspired perception system, it is necessary to combine sensors with artificial synapses and neurons. Constructing synapses and neurons with traditional CMOS technology requires complex circuits, which results in inefficiency in the overall area and energy consumption[10,11]. Recently, emerging devices, such as memristors, have been used to emulate the functionalities of synapses and neurons due to their abundant ion dynamics[12–23]. The neuromorphic perception computing system that combines sensors and synapses/neurons has proven to be capable of processing sensory information, such as tactile[24–31], visual[32–36], nociception[37,38] signals, etc. However, these artificial neurons only handle single-mode physical signals, and most of them suffer from significant cycle-to-cycle and device-to-device variations, which are significant challenges toward applications. A neuromorphic perception computing system that can handle multi-mode physical signals and have excellent uniformity is greatly desirable.

In this study, we report a calibratable artificial sensory neuron (CASN) consisting of epitaxial VO_2 memristor grown by pulsed laser deposition and a variety of coupled sensors. The high crystalline quality of epitaxial VO_2 gives rise to significantly improved cycle-to-cycle uniformity of the artificial neuron, and a calibration resistor is further introduced to optimize the device-to-device consistency between different neurons. In addition, the artificial spiking neuron is equipped with a scaling resistor to suit different types of sensors with varied resistance levels. Based on this, we demonstrate cross-sensory neuromorphic perception component that is able to encode optical, thermal, pressure, and curvature signals into spikes, showing capability in simulating biological vision, temperature, haptic, and mechanical sensation capabilities. The perception neurons are further incorporated as the input neurons of a 3-layer spiking neural network by simulation, achieving an accuracy of 90.33% on MNIST-based pressure image classification. Finally, we have utilized these neuromorphic perception components to monitor the curvatures of fingers and thereby achieved classification of hand gestures. These results demonstrate the great potential of our CASN-based neuromorphic perception system in highly efficient multi-sensory neurorobotics.

## Results

**Calibratable spiking neuron based on epitaxial VO_2.** By utilizing various senses, humans collect physical information of the external world and encode it into spikes, which are then transmitted to the cerebral cortex for perception and learning[4,9,39]. A highly efficient neuromorphic sensory system in hardware that can process a variety of physical signals is thus desirable. Figure 1 shows the comparison between the biological perception system and our spike-based artificial neuromorphic perception system. In the biological perception system, certain types of receptors (photoreceptors, thermal receptors, mechanoreceptors, etc.) and neurons convert external environmental signals into electrical spikes (Fig. 1a). The cerebral cortex then receives these spikes and responds to external stimuli. In our spike-based artificial neuromorphic perception system, we implement a calibratable artificial sensory neuron based on epitaxial VO_2 (Fig. 1b). The CASN is able to encode different types of sensory signals into electrical spikes, and these spikes can be further processed by spiking neural network (SNN). Implementation of such sensory and computing architecture is therefore important for building highly efficient multi-sensory systems.

The VO_2 film with a thickness of 20 nm was epitaxially grown on $c$-$Al_2O_3$ substrates by pulsed-laser deposition (PLD) using 308 nm XeCl excimer laser operated at an energy density of ~1 J/cm$^2$ and a repetition rate of 3 Hz. The films were deposited at 530 °C in a flowing oxygen atmosphere at oxygen pressure of 2.0 Pa. The VO_2 memristor used in this work is designed as a planar device (Fig. 2a). Figure 2b shows scanning electron microscopy (SEM) image of the device, and in Fig. 2c the channel region is enlarged, where the channel length is 400 nm and the electrode width is 1 μm (see "Methods" for the details of fabrication processes). Figure 2d, e shows the transmission electron microscopy (TEM) image of the device, while the cross-sectional scanning transmission electron microscopy (STEM) image and corresponding energy-dispersive X-ray spectroscopy (EDS) mapping of O, Au, Ti, V, Si, and Al can be seen in Supplementary Fig. 1a, along with EDS elemental line profile in the same region (Supplementary Fig. 1b). A zoomed-in view of the film shows well-ordered lattice fringes of VO_2 (Fig. 2f), and the corresponding fast Fourier transformation (Fig. 2g) once again verifies the high crystalline quality of the epitaxially grown VO_2.

The excellent crystalline quality of the epitaxially grown VO_2 plays a crucial role in achieving high uniformity in VO_2 memristor. The VO_2 memristor exhibits volatile resistive switching as can be found from its current-voltage (I–V) characteristics (Fig. 2h), where the device can change from a high resistance state (HRS) to a low resistance state (LRS) when the applied voltage exceeds a threshold voltage ($V_{th}$) of around ±1.35 V and immediately return to HRS once the applied voltage gets lower than a holding voltage ($V_{hold}$) of around ±0.85 V. Such volatile threshold switching (TS) characteristics and metal-insulator transition in VO_2 have attracted extensive attention[40,41], which has a complex mechanism involving both electronic and structural phase transitions[42]. Supplementary Fig. 2 shows the experimental results and simulated I–V curve based on the metal-insulator transition (MIT) model, where the blue points are the experimental data and the red curve is the simulation result, along with spatial heat distribution in different stages of the phase transition. As the applied voltage progressively increases (state (1) to (2)), heat is generated in the VO_2 memristor. Once the phase transition is triggered, a filament is formed through the VO_2 gap, which switches the device from HRS to LRS. The filament is expanded as the voltage increases (state (2) to state (3)). When the applied voltage is reduced, the heat dissipates, and the filament size decreases (state (3) to state (4)). When the applied voltage is below $V_{hold}$, the filament breaks down and the device eventually returns to HRS (state (4) to state (1)), as shown in Supplementary Fig. 2. More details of the model used for simulation are provided in "Methods", Supplementary Table 1, and Supplementary Note 1. The symmetrical hysteresis curve can be seen under both positive and negative biases of a voltage sweep. Transient electrical measurements show that the switching

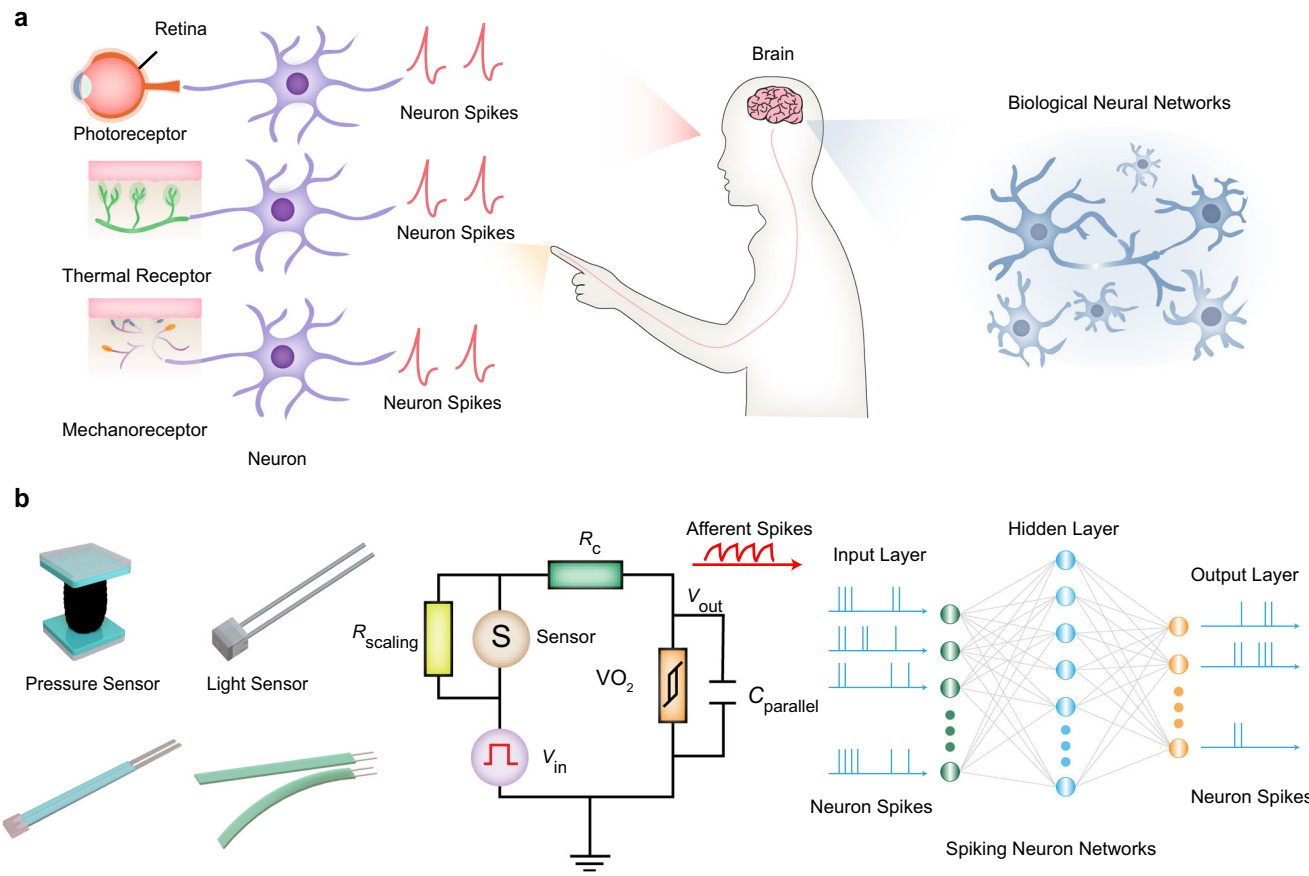

**Fig. 1 Comparison of biological perception system and spike-based neuromorphic perception system. a** Schematic of biological perception system. Biologically, physical signals from the outside world are converted by receptors and neurons into electrical impulse signals, which are then transmitted to the cortex for further processing. **b** Schematic of spike-based artificial neuromorphic perception system. The calibratable artificial sensory neuron combined with sensors (pressure sensor, light sensor, temperature sensor, curvature sensor) convert different physical signals into spikes and then transform them into the spiking neural network (SNN) for complex tasks.

speed of $VO_2$ memristor in this work is <200 ns from off-state to on-state and <75 ns from on-state to off-state (Supplementary Fig. 3 and Supplementary Note 2). Figure 2h depicts 1000 voltage sweep cycles from 0 to 2 V and 0 to −2 V, demonstrating extremely stable TS characteristics with low cycle-to-cycle (C2C) variation. Since the resistance switching in $VO_2$ is ascribed to the intrinsic electronic and structural phase transitions[42] in the material itself without necessarily incorporating defects unlike redox-based memristors, the low C2C variation can be attributed to the high crystalline quality of the epitaxial $VO_2$. The distributions of high and low resistance states of the epitaxial $VO_2$ memristor and cumulative plots of positive and negative threshold/holding voltages, including $V_{th\_pos}$, $V_{hold\_pos}$, $V_{th\_neg}$, $V_{hold\_neg}$, in 1000 repeated cycles are shown in Fig. 2i–j. Following the protocol introduced in previous studies[43], we calculated the coefficient of variation ($C_V$) as the standard deviation ($\sigma$) divided by the mean value ($\mu$). The minimum cycle-to-cycle variability in $V_{th\_pos}$, $V_{th\_neg}$, $V_{hold\_pos}$, and $V_{hold\_neg}$ was 0.73%, 0.7%, 0.51%, and 0.5%, respectively, demonstrating very low variability (Supplementary Fig. 4). The device-to-device variability in $V_{th\_pos}$, $V_{th\_neg}$, $V_{hold\_pos}$, and $V_{hold\_neg}$ was 5.32%, 5.12%, 6.96%, and 7.16%, respectively (Supplementary Fig. 5). Notably, Chen et al. have reported low C2C variability of 1.53% and low D2D variability of 5.74% in hexagonal boron nitride-based crossbar arrays[43]. Our present epitaxial $VO_2$-based memristor hence demonstrates extremely low C2C variability and reasonably low D2D variability due to its high crystalline structure.

It should be noted that despite the high film quality, PLD is still limited in preparing large-scale thin films. Many methods have been adopted to synthesize high-quality $VO_2$ films, however, the growth of wafer-scale, high-quality $VO_2$ films with excellent phase transition property is still a challenge. To date, 2-inch epitaxial $VO_2$ film grown by molecular beam epitaxy was reported[44], and preparation of large-scale $VO_2$ films by electron-beam evaporation[45], thermal oxidation[46], sol-gel method[47], and sputtering[48] has also been reported. Nevertheless, the crystalline quality of the $VO_2$ film might be compromised in some preparation processes, and the growth method should be selected based on the detailed requirements on sample scale and crystalline quality in the applications.

Here, the threshold switching characteristics of epitaxial $VO_2$ memristors are used to realize spiking neurons, and the circuit configuration is shown in Fig. 3a. The epitaxial $VO_2$ memristor is connected in parallel with a capacitor and this structure is in turn in series with a load resistor $R_L$. The oscilloscope is used to measure electrical waveforms across the $VO_2$ memristor and that coming from the power supply through channels 1 and 2, respectively. The capacitor begins to charge when a voltage is applied, and once the voltage on the capacitor exceeds $V_{th}$ the $VO_2$ memristor will switch to LRS. As a result, the artificial neuron generates a spike and the capacitor will be discharged through the on-state memristor[49]. Subsequently, the voltage on the capacitor will drop below $V_{hold}$, and thus the device will return to HRS. Such charging and discharging process can be

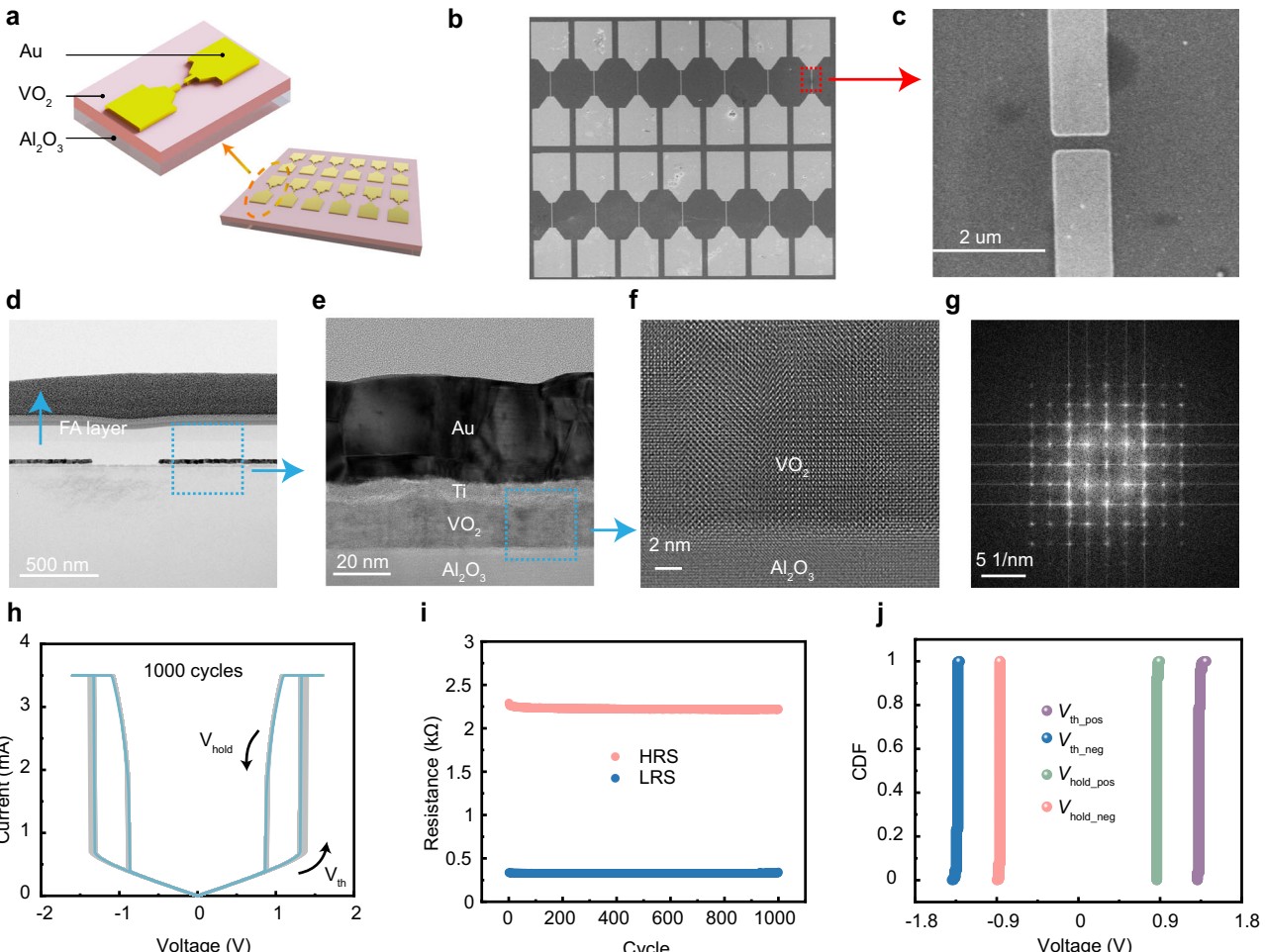

**Fig. 2 Characteristics of epitaxial VO₂ memristor. a** Schematic diagram of the memristive device, which is a planar structure. **b** Scanning electron microscopy (SEM) image of the epitaxial VO₂ memristor. **c** Zoom-in views of the channel locations in SEM. **d** Cross-sectional transmission electron microscopy (TEM) image of the epitaxial VO₂ memristor. **e** A closer view of the device in TEM. **f** Zoom-in views of the epitaxial VO₂ region. **g** The diffraction pattern extracted by fast Fourier transformation of (**f**). **h** Current-voltage characteristics of the device repeated for 1000 cycles. **i** Distributions of high and low resistance states of the epitaxial VO₂ device in 1000 repeated cycles. **j** Cumulative plots of positive threshold voltage ($V_{th\_pos}$), positive holding voltage ($V_{hold\_pos}$), negative threshold voltage ($V_{th\_neg}$), and negative holding voltage ($V_{hold\_neg}$).

clearly observed in Supplementary Fig. 6. Once the spike generation is stabilized, the charging period should occur from $V_{hold}$ to $V_{th}$, and the discharging period is from $V_{th}$ to $V_{hold}$. The spiking rate of the artificial neuron is hence affected by the series resistance, applied voltage, and parallel capacitance. Figure 3e exemplarily the spiking patterns of the artificial neuron when adopting different $R_L$ (2.6 kΩ, 5 kΩ) under a constant input voltage of 5 V without an external parallel capacitor. More results with >10 varied $R_L$ values can be found in Supplementary Fig. 7, and the spiking frequency ($f$) is summarized in Fig. 3b, showing that the frequency gradually decreases as $R_L$ increases. This is because a larger $R_L$ reduces the input current and thereby decreases the rate of charge accumulation on the capacitor. On the other hand, Fig. 3f shows the spiking waveforms with varied input voltage (4.4 V, 6.4 V) when $R_L$ is fixed as 4 kΩ, and more results can be found in Supplementary Fig. 8 and summarized in Fig. 3c. It can be seen that the spiking frequency increases as the input voltage increases, similar to biological neurons. Figure 3d and Supplementary Fig. 9 further reveal the relationship between the parallel capacitance and the spiking frequency, when the applied voltage is fixed at 5 V and $R_L$ is fixed at 4 kΩ. The spiking frequency gradually decreases as the parallel capacitance increases, since a larger capacitance results in a slower integration

process. In all of the cases, the VO₂ neuron displays excellent uniformity, which once again is based upon the high crystalline quality of epitaxial VO₂.

The spiking neuron can be modulated to a relatively low frequency (<150 Hz) when a 10 μF capacitor is adopted (Supplementary Figs. 10 and 11), whose spiking rate is at a similar level with the human nervous system, implying a great potential in the field of human-machine interaction. The COMSOL model of memristor we constructed showed excellent consistency with experimental results (Supplementary Fig. 12).

The high crystalline quality of epitaxial VO₂ has led to low C2C variations, as demonstrated in Figs. 2, 3, and Supplementary Figs. 7–11, whereas device-to-device (D2D) variations might still exist, due to fabrication imperfections, etc. We have therefore tested different artificial neurons, and their $R_L$-, voltage- and capacitance-modulation curves are shown in Fig. 3g–i. Despite the similarity of the modulation trends among the neurons, there is still considerable variation and shift among them, which still poses significant challenges toward applications. In order to further reduce the D2D variation, we have introduced a serial calibration resistor into the neuron circuit (Fig. 3j). Figure 3k demonstrates that the $f$-$R_L$ modulation curve can be well controlled by the calibration resistance $R_c$ (more experimental

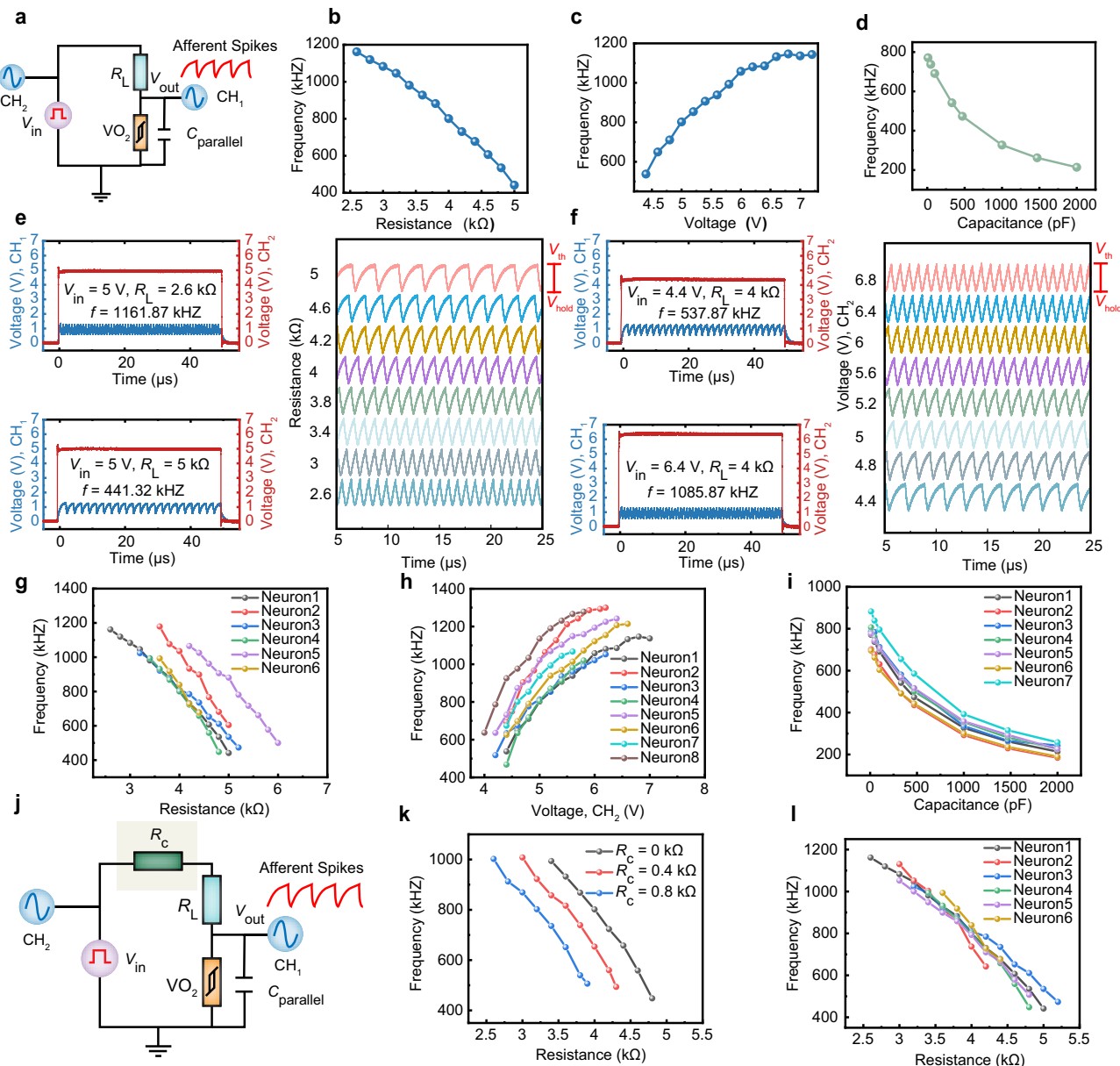

**Fig. 3 The artificial spiking neuron with its calibration design. a** Illustration of circuit based on epitaxial $VO_2$ device for implementation of a spiking neuron. **b** The effect of series resistance $R_L$ on spiking frequency. The frequency of spiking decreases as the $R_L$ increases. **c** The effect of applied voltage on spiking frequency. The firing frequency increases with the increase of the applied voltage. **d** The effect of the parallel capacitor on spiking frequency. As the parallel capacitor increases, the firing frequency gradually decreases. **e**, **f** Artificial spiking neuron response under different series resistance $R_L$ and applied voltage. **g–i** The effect of series resistance ($R_L$), applied voltage ($V_{in}$), and parallel capacitors on spiking frequency ($f$) of different neurons, respectively. Variation from neuron to neuron is easily observable. **j** The circuit structure of calibratable artificial spiking neuron. **k** The effect of series resistance $R_L$ on spiking frequency under different calibration resistances ($R_c$). **l** The relationship between the spiking frequency of different neurons and the series resistance $R_L$ after calibration. It is observed that the variation between neurons is effectively reduced compared with (**g**).

data are shown in Supplementary Fig. 13). This, therefore, offers a valuable mechanism, based on which we can shift and align all the modulation characteristics from different neurons. Indeed, experimental results demonstrate that the D2D variation has been effectively reduced (Fig. 3l) compared with Fig. 3g. The combination of epitaxial $VO_2$ and calibration resistance have therefore addressed the C2C and D2D variations, respectively, which dramatically enhanced the uniformity of the spiking neurons. The power consumption of the spiking neuron is displayed in Supplementary Fig. 14. The transient power is calculated by multiplication of input voltage with output current, and energy consumption is calculated by dividing the total energy

consumption by the spike number, which gives rise to ~2.9 nJ for each spike. This value is still lower than the state of the art reporting few pJ/spike (ref. [28]). The relatively high energy consumption originates from two main factors: the relatively low resistance and the relatively high $V_{th}$. The resistance of the device can be improved by optimizing the growth conditions of the $VO_2$ film. On the other hand, it is expected that the threshold voltage could be reduced by decreasing the channel length of the $VO_2$ memristor. To demonstrate this, we have optimized the thin film growth conditions of $VO_2$ and one can see that the current is reduced from mA level to 50–80 µA (see detailed results in Supplementary Fig. 15a–c). The resistance of the device has

increased by nearly two orders of magnitude as shown in Supplementary Fig. 15d. Moreover, significant reduction in $V_{th}$ and $V_{hold}$ could indeed be achieved by decreasing the channel length of the VO$_2$ memristor (Supplementary Fig. 15e). Future work will focus on continued optimization of the growth conditions for VO$_2$ films and scaling the size of the devices to further reduce the energy consumption.

**Spike-based neuromorphic sensory system for multi-mode perception.** Human receives different types of sensory signals from the environment through different receptors, and the signals are encoded into spikes and sent to the cerebral cortex, allowing them to learn and perceive. Based on the calibratable spiking neuron depicted above, coupled with a variety of sensors, a neuromorphic perception system for tactile, optical, and temperature perception is realized.

We have first fabricated a pressure sensor based on graphene aerogel (as shown in Supplementary Fig. 16a), which has gained wide attention due to their low density, novel electrical properties, high mechanical strength and chemical stability[50–52], and integrated it with the spiking neuron to realize tactile perception. Pressure sensors based on graphene aerogel can be easily fabricated by sandwiching a graphene aerogel layer between two thin copper electrodes and wrapping the overall structure in PVA protective film. Such sensor can have different stable resistance values under different pressures, as shown in Supplementary Fig. 16b, because in contact with the almost flat surface of the copper electrode, the graphene aerogel has a rough surface made up of many graphene flake ends. When external pressure is applied, the graphene aerogel is deformed, causing a larger number of the graphene flake ends to contact the electrodes and therefore decreasing the resistance. Supplementary Fig. 16c shows the dependence of resistance response on pressure. Application of pressure from 0 to 0.98 N has resulted in a change in the sensor resistance from 81 to 2.7 kΩ, revealing a wide range of resistance output. The output resistance effectively replaces the $R_L$ in the original neuron circuit, therefore mapping the sensory signal to the spiking frequency of the neuron based on the $f$-$R_L$ modulation (Fig. 3b, g, k, l). Indeed, Fig. 4b shows the spiking frequency of the tactile perception neuron as a function of the pressure where an external parallel capacitor (2000 pF) and a constant bias voltage (5 V) were applied, showing that the spiking frequency increases monotonously from 104.8 to 253.5 kHz when the pressure is increased from 0 to 0.98 N. The spiking response under different pressures can be observed in Fig. 4c, and more detailed results can be found in Supplementary Fig. 17. Nevertheless, the functioning of the spiking neuron places a requirement on the amplitude of $R_L$, but different types of sensors may fall into different regions of output resistance. To accommodate varied types of sensors, we have further incorporated a scaling resistor $R_{scaling}$ to adjust the resistance range to the desired range, as shown in Fig. 4a, d, g. Based on this circuit configuration, a spiking vision neuron has also been realized by introducing a light sensor (Fig. 4d). Figure 4e shows the spiking frequency of the vision perception neuron as a function of the illuminance, and the spiking response under different illuminance can be observed in Fig. 4f (more detailed results can be found in Supplementary Fig. 18). As the illuminance increases from 0 to 1275 Lux, the resistance of the light sensor decreases, resulting in a higher spiking frequency from 111.2 to 282.69 kHz (Fig. 4e, f), therefore encoding light into spike rates. Similarly, an artificial temperature sensory neuron is constructed experimentally (Fig. 4g, h, i, Supplementary Fig. 19). As the temperature increases, the resistance of the temperature sensor decreases, leading to increased spiking frequency (Fig. 4h, i). As a result, the

calibratable sensory neuron based on epitaxial VO$_2$ can emulate neuromorphic tactile, vision, and temperature perception systems, and is able to convert pressure, illuminance and temperature into rate-encoded spikes, therefore providing a potential as an interface between the external environment and neuromorphic computing systems.

Figure 4j shows the schematic of a spike-based neuromorphic sensory computing system for MNIST-based pressure image classification. Here, 784 spiking tactile sensory neurons are used to sense the pressure and encode it into pulses with different frequencies. The value of each pixel in a handwritten digit image is regarded as pressure. The encoded spike trains are then processed by a spiking neural network with three layers, which consists of 784 input neurons, 196 hidden neurons, and 10 output neurons. The pressure images can be divided into 10 different categories after training the network, and Fig. 4k shows the detailed simulation process. The training of SNN has been done online using backpropagation based on the experimentally measured electrical characteristics of VO$_2$ devices and the dependence of the spiking frequency of the artificial tactile neuron on the pressure (Supplementary Fig. 20). Detailed procedure for the simulation can be found in Supplementary Note 3. Figure 4l shows the evolution of the test accuracy during training process, where the classification accuracy on the test set can reach 90.33% after 50 training epochs. The averaged spiking numbers of neurons in the output layer after 50 training epochs are shown in Fig. 4m. The columns represent the labels of the input pressure images, while the rows describe the index of 10 output neurons and the color bars represent the average number of spikes. The input pressure image is correctly identified in most cases after 50 training epochs. Figure 4n further shows a confusion matrix of the classification results of the 10,000 test dataset after 50 epochs. The columns here designate the category of actual pressure image, while the rows show the classification results and the color bars represent the number of instances. Most of the pressure distribution pictures can be classified correctly after 50 epochs, once again showing that the spike-based neuromorphic perception system in this work is capable of converting physical signals into spikes and completing complex tasks.

To evaluate the sensing performance, the sensitivity of the spiking sensory neurons is defined as $S = \Delta f / \Delta x$, where $\Delta f$ and $\Delta x$ are the values of change in the neuronal frequency and input, respectively. The spiking sensory neuron is able to achieve high sensitivity of 151.74 kHz/N, 0.13 kHz/Lux, and 2.8 kHz/°C in tactile, optical, and temperature perception, respectively. Given that our pressure sensor is 2 cm in diameter, the calculated sensitivity to pressure is 47.67 kHz/kPa, which is slightly lower than the 60.8 kHz/kPa reported in ref. [30]. This can be improved by increasing the sensitivity of the pressure sensor itself. As for the sensitivity to temperature and light intensity, there seem to be no prior works reporting such metrics that can serve as the background for direct comparison, to the best of our knowledge. The important point is that our spiking sensory neuron can be matched with different kinds of sensors, which is a significant advantage over existing studies. Furthermore, the signal-to-noise ratio (SNR)[53] of the spiking sensory neuron is defined for the first time, which can be described as:

$$\text{SNR} = 10 \log \frac{\mu(f)^2}{\sigma(f)^2} \qquad (1)$$

where $\mu(f)$, $\sigma(f)$ are the mean and standard deviations of spiking frequency in every oscillation cycle, respectively. Statistical analysis on the experimental results shows that the artificial spiking sensory neurons can achieve SNRs of 33.66, 31.90, and 29.92 dB in the tactile, optical, and temperature sensing using our approach. These

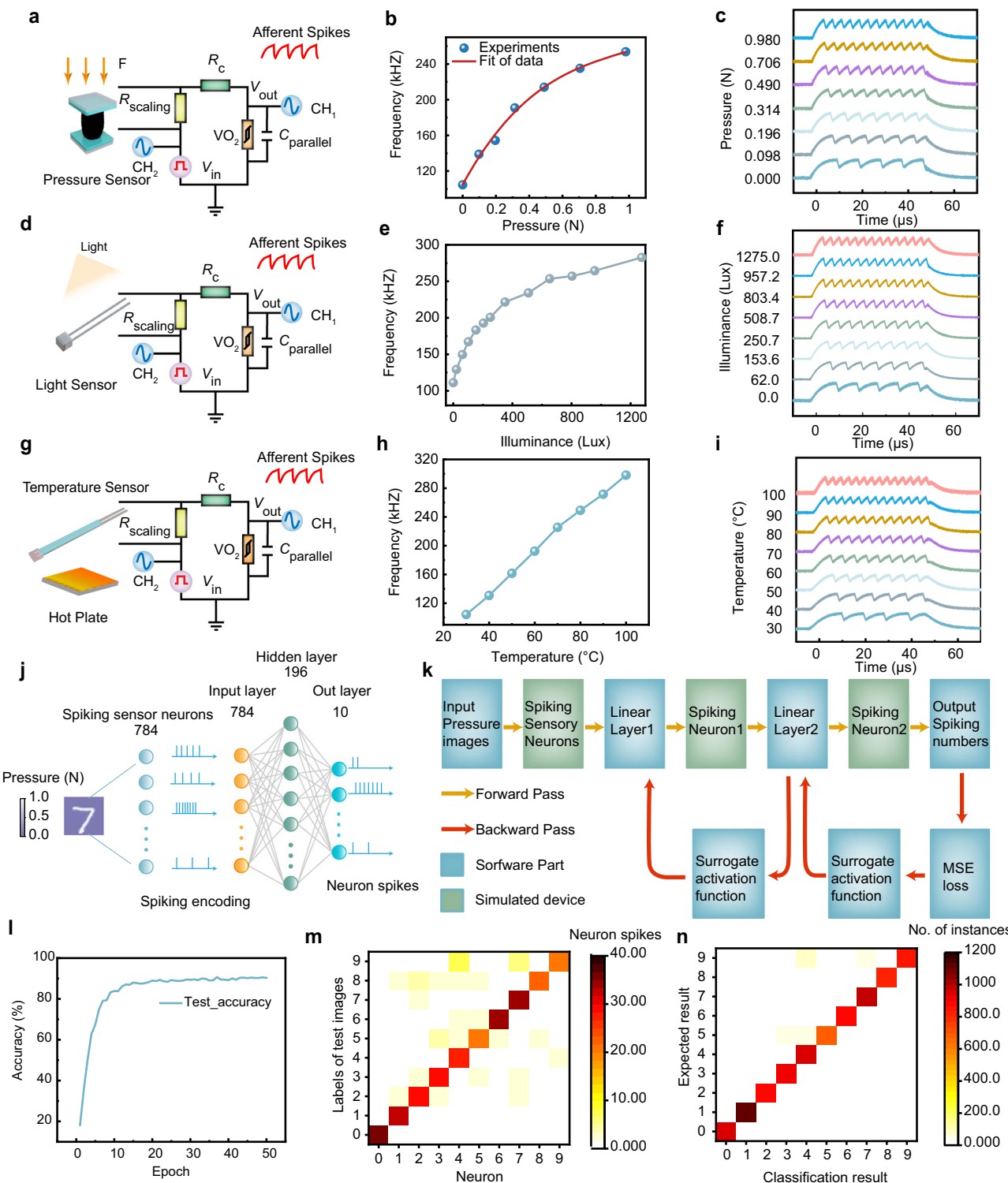

SNR values can originate from the sensor, circuit as well as fluctuations in $V_{th}$, $V_{hold}$, $R_{off}$, and $R_{on}$ of the $VO_2$ device.

Moreover, we established a model of the spiking sensory neuron. The artificial spiking neuron circuit is essentially an RC circuit. Using Kirchhoff's Current Law, we have the following differential equation:

$$C_m \frac{dV_m}{dt} = \frac{V_{in} - V_m}{R_L} - \frac{V_m}{R_{VO_2}} \qquad (2)$$

where $C_m$ is the capacitance in parallel to the $VO_2$ device or can be parasitic capacitance. $V_m$ is the output voltage across the $VO_2$ device. The $VO_2$ resistance is $R_{VO_2} = R_{off}$ in HRS and $R_{VO_2} = R_{on}$ in LRS. For simplicity, we assume that $R_{off}$ and $R_{on}$ are constant in our analyses.

To obtain the rising time, $t_r$, from $V_{hold}$ to $V_{th}$ during oscillation, we analyze the circuit when $R_{VO_2} = R_{off}$. By integrating the equation and applying the initial condition

**Fig. 4 Illustration of the spike-based neuromorphic perception system for tactile, optical, and temperature perception. a** Schematic of artificial spiking tactile sensory neuron. The graphene aerogel-based pressure sensor is combined with a calibratable artificial sensory neuron to replace $R_L$ in the original neuron circuit. A scaling resistor ($R_{scaling}$) is used to adjust the range of the sensor resistance where $R_{scaling}$ is set as 4 kΩ. **b** The effect of pressure on spiking frequency. **c** Artificial spiking tactile sensory neuron response under different pressure. **d** Schematic of artificial spiking vision sensory neuron where $R_{scaling}$ is set as 3 kΩ. **e** The effect of illuminance on spiking frequency. **f** Artificial spiking vision sensory neuron response under light intensity. **g** Schematic of the artificial spiking temperature sensory neuron where $R_{scaling}$ is set as 4.5 kΩ. **h** The effect of temperature on spiking frequency. **i** Artificial spiking temperature sensory neuron response under different temperatures. **j** Schematic of the spike-based neuromorphic perception computing system for MNIST-based pressure image classification. The value of each pixel in a handwritten digital image is regarded as pressure. A pressure image is translated into spikes by 784 artificial spiking tactile sensory neurons, which are input into a three-layer spikes neural network (SNN), and finally, we calculate the firing rate of neurons in the output layer to get the classification result. **k** Flow chart of the simulation process. In the forward process (yellow arrow), first, a pressure distribution image is encoded into spikes by the spiking sensory neurons, then sent to the linear layer for weighted, and integrated on the spiking neurons. The spiking rate of the output layer is used to calculate the loss. In the error backpropagation stage (red arrow), the sigmoid type function is used to calculate the gradient. The calculation module in the green box is simulated based on experimental data, and the calculation module in the blue box is software simulation. **l** Evolution of the test accuracy with training epochs. After 50 epochs of training, the accuracy on the test set can reach 90.33%. **m** The average spiking numbers of neurons in the output layer when different pressure images are input. **n** Confusion matrix of the classification results of the test dataset after 50 epochs showing images of pressure can be well classified.

$V_m(0) = V_{hold}$, we obtain:

$$V_m(t) = \frac{R_{off}}{R_L + R_{off}} V_{in} - \left( \frac{R_{off}}{R_L + R_{off}} V_{in} - V_{hold} \right) \exp \left( -\frac{t}{(R_L \parallel R_{off}) C_m} \right) \tag{3}$$

At $t = t_r$, $V_m(t_r) = V_{th}$. Inserting these values into the equation and further rearranging, we arrive at the expression:

$$t_r = (R_L \parallel R_{off}) C_m \cdot \ln \left( \frac{\frac{R_{off}}{R_L + R_{off}} V_{in} - V_{hold}}{\frac{R_{off}}{R_L + R_{off}} V_{in} - V_{th}} \right) \tag{4}$$

For the falling time, $t_f$, from $V_{th}$ to $V_{hold}$ during oscillation, we let $R_{VO_2} = R_{on}$. By integrating the differential equation at initial condition $V_m(0) = V_{th}$, we arrive at the following equation:

$$V_m(t) = \frac{R_{on}}{R_L + R_{on}} V_{in} - \left( \frac{R_{on}}{R_L + R_{on}} V_{in} - V_{th} \right) \exp \left( -\frac{t}{(R_L \parallel R_{on}) C_m} \right) \tag{5}$$

At $t = t_f$, $V_m(t_f) = V_{hold}$. Hence, we have:

$$t_f = (R_L \parallel R_{on}) C_m \cdot \ln \left( \frac{V_{th} - \frac{R_{on}}{R_L + R_{on}} V_{in}}{V_{hold} - \frac{R_{on}}{R_L + R_{on}} V_{in}} \right) \tag{6}$$

Thus, the oscillating frequency is:

$$f = \frac{1}{t_r + t_f} \tag{7}$$

This model is similar to the one given in refs. [54–56], which are essentially relaxation oscillators relying on volatile threshold switching devices to enable self-sustained oscillations. However, it is worthwhile noting that the detailed time constant and voltage dividing terms across the capacitor are different due to their different circuit designs, and more importantly, this model takes into account the scaling resistance and the calibration resistance. Extending the model to calibratable spiking sensory neuron, we have:

$$R_L = R_{sensor} \parallel R_{scaling} + R_c \tag{8}$$

where $R_{sensor}$, $R_{scaling}$, and $R_c$ are resistance of sensor, scaling resistance, and calibration resistance.

To evaluate potential device variation of VO$_2$ memristors on precise sensing, the SNR is calculated by our model considering the factors related to the VO$_2$ device, i.e., the distribution of $V_{th}$, $V_{hold}$, $R_{off}$, and $R_{on}$ (Fig. 2h–j), showing SNR of 36.38 dB ($V_{in}$, $R_L$, and $C_m$ are set as 5 V, 2.6 kΩ, and 1000 pF, respectively).

It is worth noting that VO$_2$ is a system that is very sensitive to oxygen content, protons, and temperature in ambient environment. In order to examine these factors, we have first performed control experiments to measure the characteristics of VO$_2$ devices under different atmospheric pressure, including air (Supplementary Fig. 21a), varied atmospheric pressure from $1.5 \times 10^{-3}$ mbar to $2 \times 10^{-4}$ mbar (Supplementary Fig. 21b–h) and N$_2$ environment (Supplementary Fig. 21i). Therefore, the concentration of oxygen and moisture/proton is gradually reduced in this process, where the VO$_2$ device showed no significant change in its $I$–$V$ characteristics. To quantify the impact, the threshold and holding voltages ($V_{th\_pos}$, $V_{th\_neg}$, $V_{hold\_pos}$, and $V_{hold\_neg}$) as well as $R_{off}$ and $R_{on}$ of the devices at different atmospheric pressures are extracted (Supplementary Fig. 21j–k). The highly stable threshold and holding voltages as well as resistance states demonstrate that the VO$_2$ memristor can operate stably under varied oxygen and moisture concentrations. To model and address the effect of temperature, we tested the $I$–$V$ characteristics of the device at different temperatures. The $I$–$V$ characteristics of VO$_2$ memristor at 283–305 K are displayed in Supplementary Fig. 22a–h. Supplementary Fig. 22i shows $V_{th}$ and $V_{hold}$ at different temperatures, where one can find that both $V_{th}$ and $V_{hold}$ gradually decrease with increased temperature. Moreover, the firing frequency of the VO$_2$ spiking neuron at different temperatures is further tested from 283 to 291 K with constant $R_L$ of 4 kΩ and the same input voltage of 5 V (Supplementary Fig. 22k), where the devices are placed directly on a temperature-controlled probe station (Supplementary Fig. 22j). As the temperature increases, one can find that the firing frequency of VO$_2$ neurons gradually increases (576.13–656.02 kHz). We have systematically tested the dependence of the spiking frequency as a function of load resistance ($R_L$) and temperature, and the results are displayed in Supplementary Fig. 22l, showing similar $R_L$ and $T$ dependence in all cases. This might be ascribed to the gradual decrease in threshold voltages of VO$_2$ memristors with increased temperature, so that the neuronal circuit requires lower voltage to fire.

It is worth noting that the relatively low phase transition temperature ($T_t$) of VO$_2$ could limit the operating temperature of neuromorphic systems and poses a challenge in electronic applications. Hence, appropriate material engineering to increase $T_t$ is highly desirable. A possible strategy to increase the $T_t$ of VO$_2$ is by doping. For example, doping by Cr$^{3+}$, Ge$^{4+}$, and Ti$^{4+}$ cations[57–60] have been reported to increase $T_t$ of VO$_2$ thin films, with Cr-doped VO$_2$ and Ge-doped VO$_2$ showing $T_t$ of ~100 °C and ~95 °C, respectively. Besides, $T_t$ may also be modulated via strain engineering. In particular, tensile strain along the $c$-axis of

the VO$_2$ lattice results in a higher $T_t$. It has been demonstrated that VO$_2$ thin films deposited on TiO$_2$(100)[61] and TiO$_2$(110)[62] substrates have increased $T_t$ due to substrate-induced strain, with the latter reporting a $T_t$ of ~95 °C. Such transition temperature range, by means of doping or strain engineering, is more favorable for practical applications.

After considering the effect of temperature, the threshold and holding voltages are corrected and can be described as[63]:

$$V_{th}(T) = \sqrt{\frac{R_{off}}{R_{th}}\left(T_t - T\right)} \qquad (9)$$

$$V_{hold}(T) = \sqrt{\frac{R_{on}}{R_{th}}\left(T_t - T\right)} \qquad (10)$$

where $R_{th}$, $T_t$, and $T$ are the effective thermal resistance, the transition temperature of VO$_2$, and the operating temperature, respectively. Therefore, the impact of temperature on VO$_2$ neuron spiking can be obtained by inserting Eqs. (9) and (10) into Eqs. (2)–(8). The validity of this model has been verified in Supplementary Fig. 23, where a set of $f$-$R_L$ curves under different temperatures are calculated using this model (the values of the parameters are shown in Supplementary Table 2). The calculated results are well consistent with the experimentally measured data (Supplementary Fig. 22l), hence demonstrating the reliability of our model.

**Spike-based neuromorphic perception system for gesture recognition**. Human gesture recognition is valuable in fields such as healthcare[64], human-machine interaction[65,66], and cognitive neuroscience research[67]. A highly efficient gesture-sensing system that works like biological systems is therefore desirable. Here, a spike-based neuromorphic perception system for gesture recognition that can encode hand gestures into differentiable spikes has been experimentally implemented (Fig. 5). First, we designed and fabricated a spiking curvature sensory neuron that can convert curvatures into spikes, which is composed of a curvature sensor attached in series with the calibratable spiking neuron (Figs. 2 and 3), as shown in Fig. 5a. When the curvature is increased, the resistance of the curvature sensor will be increased, which in turn leads to reduced spiking frequency of the neuron. Afterward, the curvature sensor is attached onto a human finger, so as to sensing the bending of the finger. One can see that the spiking frequency of the sensory neuron becomes lower when the finger is bent to a larger extent, as shown in Fig. 5b.

To achieve hand gesture recognition, the thumb, index, middle, ring, and little fingers are paired with 5 spiking curvature sensory neurons, and we experimentally monitored and measured the output spike trains of 8 different hand gestures. The corresponding spike profiles of the hand gestures are shown in Fig. 5c. Figure 5d statistically analzyed the spiking frequency of the sensory neurons in each finger under the different gestures. It can be seen that the spiking frequency encoded by each gesture can be easily distinguished, hence demonstrating that the spike-based neuromorphic perception system can be effectively used for gesture recognition.

The calibratable artificial sensory neurons based on epitaxial VO$_2$ shown here demonstrate significant advantages compared with traditional silicon circuits and other spiking sensory neurons. Supplementary Fig. 24 schematically depicts the comparison between neuromorphic perception system based on silicon circuits and our approach. In traditional silicon-based circuits, in order to sense physical signals a large number of ADCs (analog-to-digital converters) are necessary besides the sensors, which are very costly in area and energy consumption, and when the subsequent information processing is in spike-based neuromorphic computing systems, a large number of additional VSCs (voltage-to-spike converters) will be required[30] to realize spike conversion, which also consume a large amount of area and energy, as shown in Supplementary Fig. 24a. In stark contrast, our calibratable sensory neuron (Supplementary Fig. 24b) can directly achieve both sensing and spike conversion with the simple circuit consisting of the sensor, the VO$_2$ memristor, and a few resistors and capacitors, which is much more efficient in area and energy consumption compared to silicon circuits. Supplementary Table 3 further benchmarks our approach with other state-of-the-art spike-based sensory neurons. Compared with existing works in the literature, our work effectively addressed the impedance matching problem between sensors and neurons by utilizing the scaling resistance and calibration resistance to adapt the working resistance ranges of different sensors. As a result, a variety of different perception modalities including pressure, light, temperature, and curvature have been achieved for the first time, which is a significant advantage of our approach and not seen in existing studies. The high crystalline quality of epitaxial VO$_2$ has addressed the fundamental cycle-to-cycle and device-to-device variation issues in sensory neurons, and the resultant excellent uniformity of our devices gives rise to excellent SNRs of 33.66, 31.90, and 29.92 dB in tactile, optical and temperature sensing, respectively. Our investigations have revealed that the present energy consumption, sensitivity, and firing frequency of the sensory neurons could be further improved by optimizing the growth conditions of the VO$_2$ film (Supplementary Fig. 15a–d), the channel length (Supplementary Fig. 15a–e), and circuit parameters such as parallel capacitance (Fig. 3, Supplementary Figs. 7–11).

## Discussion

A highly uniform, calibratable artificial sensory neuron based on threshold switching in epitaxial VO$_2$ memristor has been experimentally implemented for the first time. The epitaxial VO$_2$ memristor has excellent cycle-to-cycle and device-to-device uniformity, due to the high crystalline quality of epitaxially grown VO$_2$ and introduction of calibration resistor. A variety of spiking sensory neurons can be constructed based on the CASN capable of sensing and converting physical signals into spikes, and a scaling resistor can be further used to accommodate varied types of sensors by adjusting their various resistance ranges to the desired regime. Based on this, a multi-sensory perception system capable of encoding pressure, curvature, illuminance, and temperature into electrical spikes is demonstrated experimentally by combining CASN with pressure, curvature, light, and temperature sensors. Simulation results show that combination of the spiking tactile neurons with a 3-layer SNN can lead to successful pattern classification on pressure images, showing classification accuracy of >90.33%. A spike-based neuromorphic perception system with spiking curvature sensory neurons has been utilized to achieve hand gesture recognition experimentally. This study could extend the currently limited sensing mode of sensory neurons and address their fundamental cycle-to-cycle and device-to-device variation issues, therefore significantly promoting the development of neurorobotics, perception, and neuromorphic computing.

## Methods

**Fabrication of epitaxial VO$_2$ threshold switching devices**. The 20 nm VO$_2$ films were epitaxially grown on $c$-Al$_2$O$_3$ substrates by pulsed-laser deposition (PLD) technique using a 308-nm XeCl excimer laser operated at an energy density of about 1 J/cm$^2$ and a repetition rate of 3 Hz. The VO$_2$ films were deposited at 530 °C

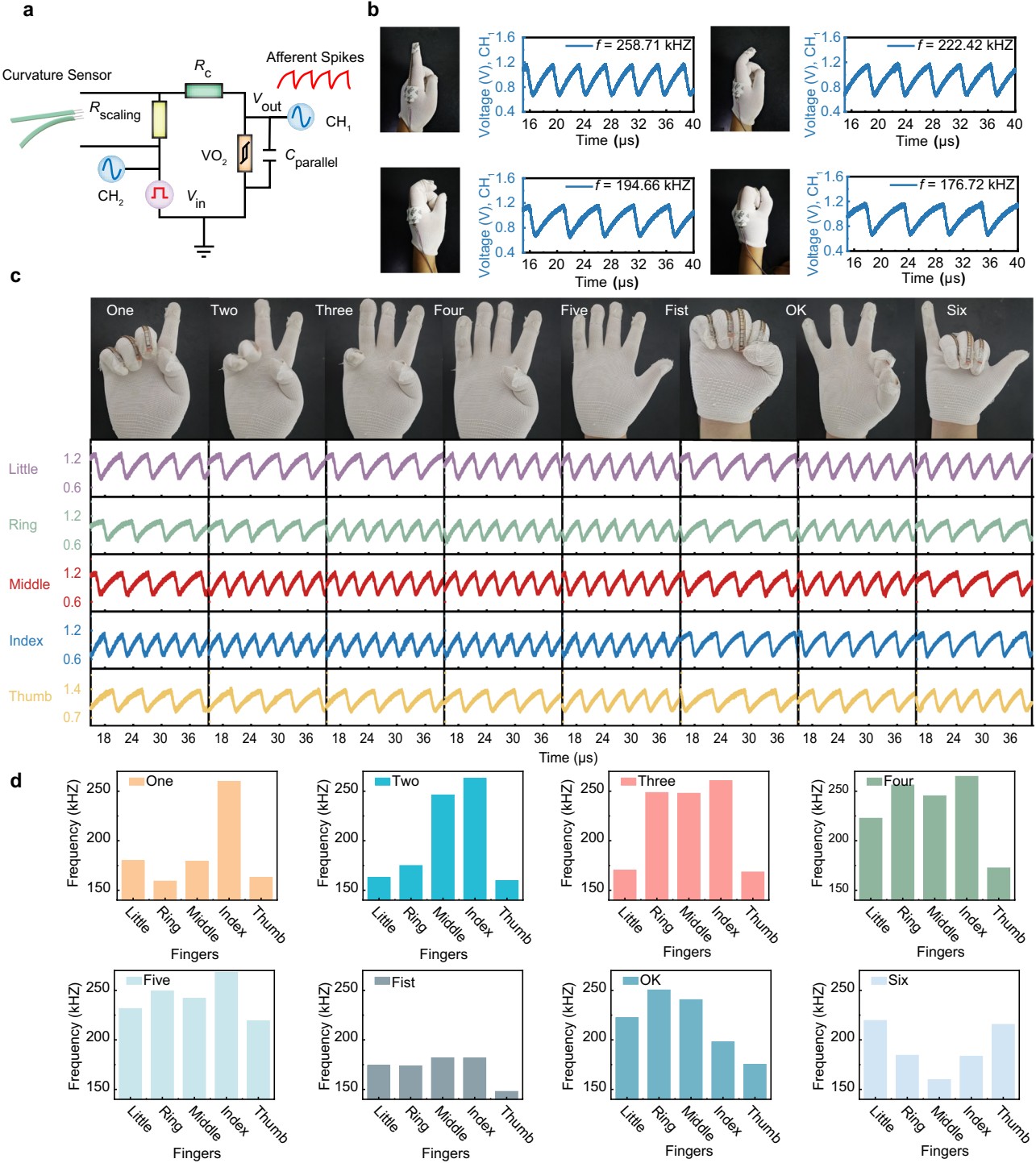

**Fig. 5 Illustration of the spike-based neuromorphic perception system for gesture recognition. a** Schematic of artificial spiking curvature sensory neuron. The curvature sensor is combined with a calibrated artificial spiking neuron, replacing the $R_L$ in the original spiking neuron circuit. **b** Artificial spiking curvature sensory neuron response under different curvatures. With the curvature sensor attached to the finger, the spiking frequency of the curvature sensory neuron depends on how much the finger bends. The greater the bend of the finger, the lower the frequency of spiking. **c** Artificial spiking curvature sensory neurons response under different gestures. A person's five fingers are attached with artificial curvature sensory neurons. The spiking frequencies of the five neurons change with the gestures, which constitute a neuromorphic perception system for gesture recognition where the gestures can be encoded into five spike trains. **d** Statistics of spiking frequency of spiking curvature sensory neurons in five fingers during different gestures showing each gesture can be easily distinguished.

in a flowing oxygen atmosphere at the oxygen pressure of 2.0 Pa. Then, the films were cooled down to the room temperature at the speed of 20 °C/min. The deposition rate of $VO_2$ thin films was calibrated by X-ray Reflection (XRR).

The electrodes, which are composed of Au (40 nm) and Ti (5 nm) with a distance of 400 nm, were patterned with electron-beam lithography (EBL) along with electron-beam evaporation and lift-off.

**Electrical measurements**. Electrical measurements were performed using an Agilent B1500A semiconductor parameter analyzer, Keithley 2450 SourceMeter, and the RIGOL MSO8104 digital storage oscilloscope. We used an Agilent B1500A semiconductor parameter analyzer to perform electrical measurements of a single $VO_2$ device in Fig. 2h–j and Supplementary Figs. 2–5. In Figs. 3–5 and Supplementary note 4–5, Agilent B1500A is applied to create the pulse signal, and one channel of the oscilloscope is used to measure the output of Agilent B1500A, while the other channel measures the voltage of the output node in the spiking neuron circuit. In Fig. 3b, the series resistance $R_L$ is changed to explore the relationship between the spiking frequency and the series resistance $R_L$ where the voltage applied is 5 V. In Fig. 3c, the series resistance $R_L$ is set at 4 kΩ, we change the input voltage to see how it relates to the spiking frequency. In Fig. 3d, the voltage is set to 5 V and the series resistance $R_L$ is set to 4 kΩ, with the parallel capacitor changed to investigate the relationship between the spiking frequency and the parallel capacitor. More experimental circuit parameters are shown in Supplementary Table 4. In the spike-based neuromorphic perception system (Figs. 4 and 5), we used the off-the-shelf light sensor (GL3537-1), temperature sensor (NTC10KB3950), and curvature sensor (FLEX4.5).

**Simulation of $VO_2$ device in COMSOL**. We simulated the operation of the $VO_2$ device in COMSOL Multiphysics software based on the metal-insulator transition (MIT) model described in ref. [68]. In this model, the resistive switching process is simply due to the temperature change in the switching region resulting from the interplay of Joule-heating and the heat dissipation of the device. The thermally-activated high resistivity of $VO_2$ in the lower temperature range is given by Eq. (11):

$$\rho_m = \rho_{0,m} \cdot \exp\left(\frac{E_{a,m}}{k_b T}\right) \tag{11}$$

where $E_{a,m}$ is the activation energy. For the low resistivity ($\rho_r$) region in the higher temperature range, we used the same equation with different values of $\rho_{0,r}$ and $E_{a,r}$. As both high and low resistivity phases coexist during the transition, the switching region can be regarded as a parallel circuit. Hence, the overall resistivity is given by Eq. (12):

$$\rho = \frac{\rho_r \rho_m}{f_r \rho_m + (1 - f_r)\rho_r} \tag{12}$$

where $f_r$ is the volume fraction of the low resistivity phase and is given by Eq. (13):

$$f_r = \frac{1}{1 + A \cdot \exp\left(\frac{W}{k_b T}\right)} \tag{13}$$

$W$ is the energy scale of the MIT and is related to the steepness of the resistivity change. $A$ is a constant related to the temperature at which the MIT takes place. The values of $A$ during the heating process ($A_h$) and the cooling process ($A_c$) are different. The parameters in the equations above were tuned so that the simulated $I–V$ curve fit the measured $I–V$ curve of our device (more details are shown in Supplementary Note 1).

**Simulation of the spike-based neuromorphic perception system**. A spike-based neuromorphic perception computing system for pressure image recognition using the artificial spiking tactile sensory neurons and spiking neural networks (SNN) is implemented in simulation by the SpikingJelly[69] based on experimental data. We used Origin to fit the pressure and spiking frequency curve in Supplementary Fig. 20a. The high resistance of the device, $V_{th}$ and $V_{hold}$ are set to 2.2 kΩ, 1.4 V, and 0.85 V, respectively, which is extracted from Supplementary Fig. 20b–c. More details are shown in Supplementary Note 3.

## Data availability

All data supporting this study and its findings are available within the article, its Supplementary Information and associated files. The source data underlying Figs. 2h–j, 3b–i, k–l, 4b, c, e, f, h, i, l–n and 5b–d have been deposited at https://zenodo.org/record/6609313#.YplfNGhBxPY or are available from the corresponding author upon reasonable request.

## Code availability

The codes used for the simulations are described in https://github.com/billyuanpku96/SNN-for-sensory-neuron or are available from the corresponding author upon reasonable request.

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

## Acknowledgements

This work was supported by the National Key R&D Program of China (2017YFA0207600 (Y.Y.)), National Natural Science Foundation of China (61925401 (Y.Y.), 92064004 (Y.Y.), 61927901 (R.H.), 92164302 (Y.Y.)), Project 2020BD010 (Y.Y.) supported by PKU-Baidu Fund, and the 111 Project (B18001 (R.H.)). Y.Y. acknowledges the support from the Fok Ying-Tong Education Foundation and the Tencent Foundation through the XPLORER PRIZE.

## Author contributions

R.Y. and Y.Y. designed the experiments. G.L. and C.G. performed the $VO_2$ film growth. Q.D. fabricated the $VO_2$ devices. Z.X. fabricated the pressure sensor. R.Y., K.Y., and C.L. performed electrical measurements. R.Y., P.J., Tiw, and Z.J. performed the simulations. R.Y. and Y.Y. prepared the manuscript. Y.Y. and R.H. directed all research. All authors analyzed the results and implications and commented on the manuscript at all stages.

## Competing interests

The authors declare no competing interests.
