## [Peer Review File · Nature Communications]

REVIEWER COMMENTS

Reviewer #1 (Remarks to the Author):

The paper is exploring a way to build spike-based neuromorphic perception system for tactile, optical, and temperature perception based on VO₂ spiking oscillators. The concept is not new but some of the reported data and potential applications are new and innovative. The paper needs an in-depth revision and many improvements before being considered for publication.

1) The authors claim that this technology can be used for neuromorphic applications where the energy consumption and, in general, the energy efficiency is a big challenge. In Fig. 2h, the characteristics of the VO₂ device repeated for 1000 cycles shows currents of the order of mAs and distributions of low and high resistive states. Two main questions arise: (1) this type of device characteristics are not meeting a low power design consideration of a neuromorphic system - how do the authors envision building neuromorphic systems with many sensing components if a single device has such a high current (and power) consumption, (2) how the distributions of the low and high states are taken into account and modeled in the design of the sensory system? Can the authors explain the limits resulting in precise sensing (what is the SNR of your sensors?) from these stochastic distributions? Without an appropriate understanding and modeling of these distributions, the use of the sensor signals could be problematic even in spiking neuromorphic sensing.

2) All the VO₂ proposed spiking structures (including the ones for light and pressure) will have a high cross-sensitivity to temperature. How do the authors propose to address in a neuromorphic sensory application the intrinsic sensitivity of IMT and MIT transitions of VO₂ to temperature? Such calibration and/or modeling is not well explained in the paper. Moreover, the operating temperature of the proposed sensors can be highly limited by the transition temperature of the VO₂ (close to 68°C), which is a big limitation and concern for many electronic applications.

3) The paper reports a series of experimental results without in-depth results of the sensing performance and without any basic modeling or discussion of what are the limits of these sensing principles versus existing state-of-the-art sensory systems for light, pressure and temperature. The paper needs to compare and discuss the achieved performance of the proposed VO₂ sensors with state-of-the-art sensor references in terms of achieved sensitivity, selectivity, SNR and power consumption. Without such type of benchmarking and discussion of the real advantages of the proposed sensing solutions the work cannot be published. Also the reference list should be updated with more recent publications of other groups, including spiking VO₂ sensors.

Reviewer #2 (Remarks to the Author):

The manuscript by R. Yuan reports on capabilities of neuromorphic sensory system on reproducing biological functions incorporating different excitation stimuli such as temperature, light and pressure. The sensory neurons were based on VO₂ epitaxial films prepared by PLD.

The shown applications are nicely demonstrating the capability of the system for emulating more complex neural functionalities.

The manuscript is well structured and written, the topic is suitable for the journal. I would recommend publication provided the comments below are appropriately addressed.

Technical Comments

1. VO₂ is a system that is very sensitive to both temperature and oxygen content. Deviations in these parameters may lead to a change in the crystal structure and/or change in the stoichiometry. Both factors lead to a change in the electronic and defect-chemical structure. Did the authors consider these specifics of VO₂?

2. In same context – properties of VO₂ can also be changed by local environment and/or protons. Was this factor considered by the experiments?

3. The authors are pointing out that VO₂ is showing metal-insulator transition. Within the field of RRAMS it is known that Mott-transition is meant but the more general audience of Nature Communications would be probably confused. I would recommend adding a short paragraph explaining the physics of the resistance change in VO₂.

4. PLD is a very suitable method for preparing high quality films in lab conditions. However, it is limited for use for applications as it cannot cover homogeneously larger surfaces (in the general case it can reliably cover about 2x2 cm² substrate). It would be good to discuss what other methods can be used for larger samples.

5. From the title and the first part of the article I have had the impression that VO₂ is used as both neuron and sensor in same time. Later in the article it becomes clear that different sensors are used in a combination with the VO₂ memristor. I would recommend presenting this at earlier stage.

6. On page 8 is written: "...variation, which can be attributed to the high crystalline quality of the epitaxial VO₂..."

It would be good to provide a short argumentation (or citation) how the good crystallinity is related to the low C2C variation.

Minor issue

- On page 5 is written: "...physical signals into electrical signals..."

The expression is not precise. Electricity is also a physical phenomenon. Therefore an electrical signals is a kind of physical signal.

Reviewer #3 (Remarks to the Author):

This article analyses a very timely and important topic. The model proposed for the artificial spiking sensory neuron seems to be coherent, and the functioning of the VO₂ device is critical for the correct functioning of the whole circuit and neural network.

The devices have a planar structure in the form of a cap. While this is different from most devices with vertical structure, the lateral size of the devices is still enough small to ensure a reasonably high integration density. In fact, for me it is very surprising that the authors achieve such a low variability in planar devices. That is very meritorious and in my opinion it is related to the high crystallinity of the VO₂ film. I may recommend the authors to quantify the variability by calculating the coefficient of variance of the VSET and VRESET, indicating the number of devices and cycles per device, and compare the obtained results with the available literature, as done in Chen et al. Nature Electronics 638–645 (2020).

Also, the controllability of the frequency of the spikes generated is very good, the authors demonstrate to have full control, and the reproducibility seems to be excellent. The experiments with temperature, light and pressure are an added value potentiating the application of these systems in different applications.

The changes here mentioned are optional, as the manuscript is really good. It is really difficult to achieve such low variability in planar devices. I believe a large amount of people would be interested on this paper. Hence, I think Nature Communications would do well publishing it.

MS No: NCOMMS-22-05908-T

Title: A Calibratable Sensory Neuron Based on Epitaxial VO₂ for Spike-based Neuromorphic Multisensory System

Response to the editor and the reviewers

We would like to sincerely thank the editor for the kind consideration of our manuscript and the reviewers for their constructive suggestions which are very valuable in improving our manuscript. We have carefully considered all the reviewers' questions and made corresponding revisions. Additional experiments, modeling, and simulations (Supplementary Fig. 2, 4, 5, 14, 15, 21-24, Equation 1-10) are performed. We have also improved the writing and presentation of the manuscript. We hope the editor and reviewers will find the revised manuscript suitable for publication in *Nature communications*. The point-to-point responses and changes made are listed below.

Comments from Reviewer #1

Overall Remarks: The paper is exploring a way to build spike-based neuromorphic perception system for tactile, optical, and temperature perception based on VO₂ spiking oscillators. The concept is not new but some of the reported data and potential applications are new and innovative. The paper needs an in-depth revision and many improvements before being considered for publication.

Our response: We would like to sincerely thank the reviewer for the very detailed and constructive suggestions. In this revised manuscript, we have carefully considered all

the points and performed new experiments, models, and simulations (Supplementary Fig. 14, 15, 22-24, Equation 1-10). Our detailed responses to the comments and corresponding changes are shown as follows.

1) The authors claim that this technology can be used for neuromorphic applications where the energy consumption and, in general, the energy efficiency is a big challenge. In Fig. 2h, the characteristics of the VO₂ device repeated for 1000 cycles shows currents of the order of mAs and distributions of low and high resistive states. Two main questions arise: (1) this type of device characteristics are not meeting a low power design consideration of a neuromorphic system - how do the authors envision building neuromorphic systems with many sensing components if a single device has such a high current (and power) consumption, (2) how do the distributions of the low and high states are taken into account and modeled in the design of the sensory system? Can the authors explain the limits resulting in precise sensing (what is the SNR of your sensors?) from these stochastic distributions? Without an appropriate understanding and modeling of these distributions, the use of the sensor signals could be problematic even in spiking neuromorphic sensing.

Our response: We would like to thank the reviewer for the constructive comments.

The first part of the question is regarding the power consumption. We agree that power consumption is very important for neuromorphic systems. In light of this advice,

we have calculated the energy consumption for each spike signal in Supplementary Fig. 14, and the energy consumption for each spike is ~ 2.9 nJ, where transient power is calculated by multiplication of input voltage with output current and the energy consumption is calculated by dividing the total energy consumption by the spike number. The relatively high energy consumption originates from two main factors: the relatively low resistance and the relatively high V_{th} .

On the one hand, the resistance of the device can be improved by optimizing the growth conditions of the VO₂ film. To demonstrate this, we have optimized the thin film growth conditions of VO₂ and fabricated new devices. The I - V characteristic curves of the devices with different channel lengths are shown in Supplementary Fig. 15a-c. One can see that the current is reduced from mA level to 50-80 μ A by optimizing film growth conditions. The resistance of the device has increased by nearly two orders of magnitude as shown in Supplementary Fig. 15d, which means that the power consumption of the device will also drop by nearly two orders of magnitude.

On the other hand, V_{th} and V_{hold} could be reduced by decreasing the channel length of the VO₂ memristor as shown in Supplementary Fig. 15e. It can be seen that when the electrode width of the device is constant (1 μ m), the threshold voltage indeed decreases when the channel length is decreased. We believe the power consumption could be further reduced by using a VO₂ device with a lower threshold voltage.

To address the question, the new results are now included as Supplementary Fig. 14 and 15 in the revised manuscript, along with the following discussion in Page 13 of the main text: *“The power consumption of the spiking neuron is displayed in*

Supplementary Fig. 14. The transient power is calculated by multiplication of input voltage with output current, and energy consumption is calculated by dividing the total energy consumption by the spike number, which gives rise to ~2.9 nJ for each spike. The relatively high energy consumption originates from two main factors: the relatively low resistance and the relatively high V_{th} . Therefore, increasing the device resistance or reducing the threshold voltage can reduce the energy consumption. The resistance of the device can be improved by optimizing the growth conditions of the VO₂ film. On the other hand, it is expected that the threshold voltage could be reduced by decreasing the channel length of the VO₂ memristor. To demonstrate this, we have optimized the thin film growth conditions of VO₂ and one can see that the current is reduced from mA level to 50-80 μ A (see detailed results in Supplementary Fig. 15a-c). The resistance of the device has increased by nearly two orders of magnitude as shown in Supplementary Fig. 15d. Moreover, significant reduction in V_{th} and V_{hold} could indeed be achieved by decreasing the channel length of the VO₂ memristor (Supplementary Fig. 15e). Both contribute to reduced power consumption for neuromorphic systems..”

Supplementary Figure 14. The energy consumption of spiking neuron. Transient power is calculated by multiplication of input voltage by output current, and energy consumption is calculated by dividing the total consumption by the spike number.

Supplementary Figure 15. The characteristics of the optimized VO₂ memristor. (a-

c) The I - V characteristics of devices with different channel lengths. **(d)** Resistance of devices with different channel lengths. **(e)** V_{th} and V_{hold} for devices with different channel lengths.

The second part of the question is how the distributions of the low and high resistance states are taken into account and modeled in the design of the sensory systems. To address this question, we now build a model of the spiking sensory neuron (Equation 2-8). Besides, we also defined the SNR (Equation 1) of spiking sensory neuron and discussed the limits in precise sensing, by calculating the SNR considering the stochastic distributions of threshold/holding voltages, R_{off} and R_{on} of the device using our model.

The artificial spiking neuron circuit is essentially an RC circuit. Using Kirchhoff's Current Law, we have the following differential equation:

$$C_m \frac{dV_m}{dt} = \frac{V_{in} - V_m}{R_L} - \frac{V_m}{R_{VO_2}} \quad (R1)$$

where C_m is the capacitance in parallel to the VO₂ device or can be a parasitic capacitance. V_m is the output voltage across the VO₂ device. The VO₂ resistance is $R_{VO_2} = R_{off}$ in HRS and $R_{VO_2} = R_{on}$ in LRS. For simplicity, we assume that R_{off} and R_{on} are constant in our analyses.

To obtain the rising time, t_r , from V_{hold} to V_{th} during oscillation, we analyze the circuit when $R_{VO_2} = R_{off}$. By integrating the equation and applying the initial condition $V_m(0) = V_{hold}$, we obtain:

$$V_m(t) = \frac{R_{off}}{R_L + R_{off}} V_{in} - \left(\frac{R_{off}}{R_L + R_{off}} V_{in} - V_{hold} \right) \exp\left(-\frac{t}{(R_L \parallel R_{off})C_m} \right) \quad (R2)$$

At $t = t_r$, $V_m(t_r) = V_{th}$. Inserting these values into the equation and further rearranging, we arrive at the expression:

$$t_r = (R_L \parallel R_{off})C_m \cdot \ln \left(\frac{\frac{R_{off}}{R_L + R_{off}} V_{in} - V_{hold}}{\frac{R_{off}}{R_L + R_{off}} V_{in} - V_{th}} \right) \quad (R3)$$

For the falling time, t_f , from V_{th} to V_{hold} during oscillation, we let $R_{VO_2} = R_{on}$. By integrating the differential equation at initial condition $V_m(0) = V_{th}$, we arrive at the following equation:

$$V_m(t) = \frac{R_{on}}{R_L + R_{on}} V_{in} - \left(\frac{R_{on}}{R_L + R_{on}} V_{in} - V_{th} \right) \exp \left(-\frac{t}{(R_L \parallel R_{on})C_m} \right) \quad (R4)$$

At $t = t_f$, $V_m(t_f) = V_{hold}$. Hence, we have:

$$t_f = (R_L \parallel R_{on})C_m \cdot \ln \left(\frac{V_{th} - \frac{R_{on}}{R_L + R_{on}} V_{in}}{V_{hold} - \frac{R_{on}}{R_L + R_{on}} V_{in}} \right) \quad (R5)$$

Thus, we can calculate the oscillating frequency:

$$f = \frac{1}{t_r + t_f} \quad (R6)$$

This model is similar to the one given in Ref. R1, R2. Extending the model to calibratable spiking sensory neuron, we have:

$$R_L = R_{sensor} \parallel R_{scaling} + R_c \quad (R7)$$

where R_{sensor} , $R_{scaling}$, and R_c are resistance of sensor, scaling resistance and calibration resistance.

As can be seen from the formula, the frequency of neuron firing is affected by V_{th} , V_{hold} , R_{off} and R_{on} of the device. Potential fluctuations in V_{th} , V_{hold} , R_{off} and R_{on} will cause the fluctuation of frequency which will limits the sensing performance. To

better analyze the limits, we defined the SNR^{R3} of the spiking sensory neuron, which can be described as:

$$SNR = 10 \log \frac{\mu(f)^2}{\sigma(f)^2} \quad (R8)$$

where $\mu(f)$, $\sigma(f)$ are the mean and standard deviations of spiking frequency in every oscillation cycle, respectively. Statistical analysis on the experimental results shows that the artificial spiking sensory neurons can achieve SNRs of 33.66 dB, 31.90 dB and 29.92 dB in the tactile, optical and temperature sensing using our approach. These SNR values can be affected by the sensor, circuit as well as fluctuation in V_{th} , V_{hold} , R_{off} and R_{on} of device. Considering the factors related to the VO₂ device, i.e. the distribution of V_{th} , V_{hold} , R_{off} and R_{on} in 1000 cycles (Fig 2h-j), the SNR is calculated using our model, which is 36.38 dB (V_{in} , R_L , C_m are set as 5 V, 2.6 kΩ, and 1000 pF respectively).

To address this question, we have added the following sentences into Page 18-20 of the revised manuscript:

“Furthermore, the signal-to-noise ratio (SNR)⁵³ of the spiking sensory neuron is defined for the first time, which can be described as:

$$SNR = 10 \log \frac{\mu(f)^2}{\sigma(f)^2} \quad (1)$$

where $\mu(f)$, $\sigma(f)$ are the mean and standard deviations of spiking frequency in every oscillation cycle, respectively. Statistical analysis on the experimental results shows that the artificial spiking sensory neurons can achieve SNRs of 33.66 dB, 31.90 dB and 29.92 dB in the tactile, optical and temperature sensing using our approach. These SNR

values can originate from the sensor, circuit as well as fluctuations in V_{th} , V_{hold} , R_{off} and R_{on} of the VO_2 device.

Moreover, we established a model of the spiking sensory neuron. The artificial spiking neuron circuit is essentially an RC circuit. Using Kirchhoff's Current Law, we have the following differential equation:

$$C_m \frac{dV_m}{dt} = \frac{V_{in} - V_m}{R_L} - \frac{V_m}{R_{VO_2}} \quad (2)$$

where C_m is the capacitance in parallel to the VO_2 device or can be parasitic capacitance. V_m is the output voltage across the VO_2 device. The VO_2 resistance is $R_{VO_2} = R_{off}$ in HRS and $R_{VO_2} = R_{on}$ in LRS. For simplicity, we assume that R_{off} and R_{on} are constant in our analyses.

To obtain the rising time, t_r , from V_{hold} to V_{th} during oscillation, we analyze the circuit when $R_{VO_2} = R_{off}$. By integrating the equation and applying the initial condition $V_m(0) = V_{hold}$, we obtain:

$$V_m(t) = \frac{R_{off}}{R_L + R_{off}} V_{in} - \left(\frac{R_{off}}{R_L + R_{off}} V_{in} - V_{hold} \right) \exp\left(-\frac{t}{(R_L \parallel R_{off})C_m} \right) \quad (3)$$

At $t = t_r$, $V_m(t_r) = V_{th}$. Inserting these values into the equation and further rearranging, we arrive at the expression:

$$t_r = (R_L \parallel R_{off})C_m \cdot \ln \left(\frac{\frac{R_{off}}{R_L + R_{off}} V_{in} - V_{hold}}{\frac{R_{off}}{R_L + R_{off}} V_{in} - V_{th}} \right) \quad (4)$$

For the falling time, t_f , from V_{th} to V_{hold} during oscillation, we let $R_{VO_2} = R_{on}$. By integrating the differential equation at initial condition $V_m(0) = V_{th}$, we arrive at the following equation:

$$V_m(t) = \frac{R_{on}}{R_L + R_{on}} V_{in} - \left(\frac{R_{on}}{R_L + R_{on}} V_{in} - V_{th} \right) \exp\left(-\frac{t}{(R_L \parallel R_{on})C_m}\right) \quad (5)$$

At $t = t_f$, $V_m(t_f) = V_{hold}$. Hence, we have:

$$t_f = (R_L \parallel R_{on})C_m \cdot \ln\left(\frac{V_{th} - \frac{R_{on}}{R_L + R_{on}} V_{in}}{V_{hold} - \frac{R_{on}}{R_L + R_{on}} V_{in}}\right) \quad (6)$$

Thus, the oscillating frequency is:

$$f = \frac{1}{t_r + t_f} \quad (7)$$

This model is similar to the one given in Ref.54,55. Extending the model to calibratable spiking sensory neuron, we have:

$$R_L = R_{sensor} \parallel R_{scaling} + R_c \quad (8)$$

where R_{sensor} , $R_{scaling}$, and R_c are resistance of sensor, scaling resistance and calibration resistance.

To evaluate potential device variation of VO₂ memristors on precise sensing, the SNR is calculated by our model considering the factors related to the VO₂ device, i.e. the distribution of V_{th} , V_{hold} , R_{off} and R_{on} (Fig 2h-j), showing SNR of 36.38 dB (V_{in} , R_L , C_m are set as 5 V, 2.6 kΩ, and 1000 pF, respectively).”

2) All the VO₂ proposed spiking structures (including the ones for light and pressure) will have a high cross-sensitivity to temperature. How the authors propose to address in a neuromorphic sensory application the intrinsic sensitivity of IMT and MIT transitions of VO₂ to temperature? Such calibration and modeling is not well explained in the paper. Moreover, the operating temperature of the proposed sensors can be highly limited by the transition temperature of the

VO₂ (close to 68°C), which is a big limitation and concern for many electronic applications.

Our response: We would like to greatly thank the reviewer for the constructive comment. Indeed, the VO₂ based spiking system is intrinsically sensitive to temperature. To model and address the effect of temperature on VO₂ devices and neurons, we have performed new experiments and additional modeling studies (Supplementary Fig. 22-23, Equation 9-10).

Firstly, we tested the I - V characteristics of the device at different temperatures, and explored the impact of temperature on the threshold/holding voltage as well as the firing frequency of VO₂ neurons (Supplementary Fig. 22). One can see from Supplementary Fig. 22a-h that the voltage window of V_{th} - V_{hold} shrinks gradually as temperature increases from 283 to 305 K, which can be more evidently seen in Supplementary Fig. 22i. Both V_{th} and V_{hold} gradually decrease with increased temperature. Moreover, the firing frequency of the VO₂ spiking neuron at different temperatures is further tested from 283 to 291 K with constant R_L of 4 k Ω and the same input voltage of 5 V (Supplementary Fig. 22k), where the devices are placed directly on a temperature-controlled probe station (Supplementary Fig. 22j). As the temperature increases, one can find that the firing frequency of VO₂ neurons gradually increases (576.13–656.02 kHz). We have systematically tested the dependence of the spiking frequency as a function of load resistance (R_L) and temperature, and the results are displayed in Supplementary Fig. 22l, showing similar R_L and T dependence in all cases. This might be ascribed to the gradual decrease in threshold voltages of VO₂ memristors with

increased temperature, so that the neuronal circuit requires lower voltage to fire.

We have further incorporated such impact of temperature into the sensory neuron model described in Equation 2-8. Since V_{th} and V_{hold} are temperature-dependent parameters (Supplementary Fig. 22i), the threshold/holding voltages are corrected and can be described as^{R4}:

$$V_{th}(T) = \sqrt{\frac{R_{off}}{R_{th}}(T_t - T)} \quad (R9)$$

$$V_{hold}(T) = \sqrt{\frac{R_{on}}{R_{th}}(T_t - T)} \quad (R10)$$

These relations were derived from the heat equation using lumped analysis where R_{th} , T_t , and T are the effective thermal resistance, the transition temperature of VO₂ and the operating temperature, respectively.

Therefore, the complete spiking sensory neuron model after considering the effect of temperature can be obtained by inserting Equation R9-R10 into Equation R1-R7. The validity of this model has been verified in Supplementary Fig. 23, where a set of f - R_L curves under different temperatures are calculated using this model (the values of the parameters are shown in Supplementary Table 2). The calculated results are well consistent with the experimentally measured data (Supplementary Fig. 22i), hence demonstrating the reliability of our model. This can therefore be used to model and assess the impact of temperature in sensing applications.

To address this question, we have added the new experimental and modeling results in Supplementary Fig. 22-23 and added the following discussion into the revised manuscript:

- Page 19-20:

“Moreover, we established a model of spiking sensory neuron. The artificial spiking neuron circuit is essentially an RC circuit. Using Kirchhoff's Current Law, we have the following differential equation:

$$C_m \frac{dV_m}{dt} = \frac{V_{in} - V_m}{R_L} - \frac{V_m}{R_{VO_2}} \quad (2)$$

where C_m is the capacitance in parallel to the VO_2 device and can be parasitic capacitance. V_m is the output voltage across the VO_2 device. The VO_2 resistance is $R_{VO_2} = R_{off}$ in HRS and $R_{VO_2} = R_{on}$ in LRS. In reality, R_{off} and R_{on} is dependent on temperature, which is in turn dependent on current. Hence, these values dynamically evolve depending on the state of the circuit. For simplicity, we assume that these values are constant in our analyses.

To obtain the rising time, t_r , from V_{hold} to V_{th} during oscillation, we analyze the circuit when $R_{VO_2} = R_{off}$. By integrating the equation and applying the initial condition $V_m(0) = V_{hold}$, we obtain:

$$V_m(t) = \frac{R_{off}}{R_L + R_{off}} V_{in} - \left(\frac{R_{off}}{R_L + R_{off}} V_{in} - V_{hold} \right) \exp\left(-\frac{t}{(R_L \parallel R_{off})C_m} \right) \quad (3)$$

At $t = t_r$, $V_m(t_r) = V_{th}$. Plugging these values into the equation and further rearranging, we arrive at the expression:

$$t_r = (R_L \parallel R_{off})C_m \cdot \ln\left(\frac{\frac{R_{off}}{R_L + R_{off}} V_{in} - V_{hold}}{\frac{R_{off}}{R_L + R_{off}} V_{in} - V_{th}} \right) \quad (4)$$

For the falling time, t_f , from V_{th} to V_{hold} during oscillation, we let $R_{VO_2} = R_{on}$. By integrating the differential equation at initial condition $V_m(0) = V_{th}$, we arrive at the

following equation:

$$V_m(t) = \frac{R_{on}}{R_L + R_{on}} V_{in} - \left(\frac{R_{on}}{R_L + R_{on}} V_{in} - V_{th} \right) \exp\left(-\frac{t}{(R_L \parallel R_{on})C_m}\right) \quad (5)$$

At $t = t_f$, $V_m(t_f) = V_{hold}$. Hence, we have:

$$t_f = (R_L \parallel R_{on})C_m \cdot \ln\left(\frac{V_{th} - \frac{R_{on}}{R_L + R_{on}} V_{in}}{V_{hold} - \frac{R_{on}}{R_L + R_{on}} V_{in}}\right) \quad (6)$$

Thus, oscillating frequency could be:

$$f = \frac{1}{t_r + t_f} \quad (7)$$

This model is similar to the one given in Ref. 54,55. Extending to calibratable spiking sensory neuron, we have:

$$R_L = R_{sensor} \parallel R_{scaling} + R_c \quad (8)$$

where R_{sensor} , $R_{scaling}$, and R_c are resistance of sensor, scaling resistance and calibration resistance.”

- Page 21:

“To model and address the effect of temperature, we tested the I-V characteristics of the device at different temperatures. The I-V characteristics of VO₂ memristor at 283-305 K are displayed in Supplementary Fig. 22a-h. Supplementary Fig. 22i shows V_{th} and V_{hold} at different temperatures, where one can find that both V_{th} and V_{hold} gradually decrease with increased temperature. Moreover, the firing frequency of the VO₂ spiking neuron at different temperatures is further tested from 283 to 291 K with constant R_L of 4 k Ω and the same input voltage of 5 V (Supplementary Fig. 22k), where the devices are placed directly on a temperature-controlled probe station

(Supplementary Fig. 22j). As the temperature increases, one can find that the firing frequency of VO₂ neurons gradually increases (576.13–656.02 kHz). We have systematically tested the dependence of the spiking frequency as a function of load resistance (R_L) and temperature, and the results are displayed in Supplementary Fig. 22l, showing similar R_L and T dependence in all cases. This might be ascribed to the gradual decrease in threshold voltages of VO₂ memristors with increased temperature, so that the neuronal circuit requires lower voltage to fire.”

- Page 22:

“After considering the effect of temperature, the threshold and holding voltages are corrected and can be described as⁶²:

$$V_{th}(T) = \sqrt{\frac{R_{off}}{R_{th}}(T_t - T)} \quad (9)$$

$$V_{hold}(T) = \sqrt{\frac{R_{on}}{R_{th}}(T_t - T)} \quad (10)$$

where R_{th} , T_t , and T are the effective thermal resistance, the transition temperature of VO₂ and the operating temperature respectively. Therefore, the impact of temperature on VO₂ neuron spiking can be obtained by inserting Equation 9-10 into Equation 2-8. The validity of this model has been verified in Supplementary Fig. 23, where a set of f - R_L curves under different temperatures are calculated using this model (the values of the parameters are shown in Supplementary Table 2). The calculated results are well consistent with the experimentally measured data (Supplementary Fig. 22l), hence demonstrating the reliability of our model.”

Supplementary Figure 22. The characteristics of the VO₂ memristor under different temperatures. (a)-(h) The I - V characteristics of devices with different temperature. The window of V_{th} - V_{hold} shrinks gradually with increasing temperature. **(i)** The V_{th} and V_{hold} at different temperatures. Both V_{th} and V_{hold} gradually decreases with increasing temperature. **(j)** Diagram of equipment to test firing frequency of

spiking neuron at different temperatures. Devices are placed directly on a temperature-controlled platform. **(k)** The firing frequency of neurons at different temperatures. With increasing temperature, the firing frequency gradually increased where input voltage is 5V and R_L is 4 k Ω . **(i)** Spiking frequency as a function of load resistance (R_L) with different temperatures.

Supplementary Figure 23. Spiking frequency as a function of load resistance (R_L) with different temperatures calculated by the model.

Supplementary Table 2. The values of the parameters used in the modeling of spiking neuron considering the temperature.

Parameter	Value	Units
R_{off}	3100	Ω
R_{on}	450	Ω
R_{th}	31313	K/W
T_t	318.736	K
V_{in}	5	V

The second part of the question is regarding the relatively low phase transition temperature T_t of VO₂, which could limit the operating temperature of neuromorphic systems and poses a challenge in electronic applications. Hence, appropriate material engineering to increase T_t is highly desirable. A possible strategy to increase the T_t of VO₂ is by doping. For example, doping by Cr³⁺ (Refs. R5-R6), Ge⁴⁺ (Ref. R7) and Ti⁴⁺ (Ref. R8) cations have been reported to increase T_t of VO₂ thin films, with Cr-doped VO₂ and Ge-doped VO₂ showing T_t of ~100 °C and ~95 °C, respectively. Besides, T_t may also be modulated via strain engineering. In particular, tensile strain along the c -axis of the VO₂ lattice results in a higher T_t . It has been demonstrated that VO₂ thin films deposited on TiO₂(100)^{R9} and TiO₂(110)^{R10} substrates have increased T_t due to substrate-induced strain, with the latter reporting a T_t of ~95 °C. Such transition temperature range, by means of doping or strain engineering, is more favorable for practical applications. Lastly, when a high operation temperature is critical for the applications, it might be possible to replace VO₂ with NbO₂, which is another metal-insulator transition material with similar physical mechanisms but a much higher phase transition temperature of ~807°C(Ref. R11).

To address this question in the revised manuscript, we have added the following discussions in Page 21: *“It is worth noting that the relatively low phase transition temperature (T_t) of VO₂ could limit the operating temperature of neuromorphic systems and poses a challenge in electronic applications. Hence, appropriate material engineering to increase T_t is highly desirable. A possible strategy to increase the T_t of*

VO₂ is by doping. For example, doping by Cr³⁺, Ge⁴⁺ and Ti⁴⁺ cations⁵⁶⁻⁵⁹ have been reported to increase T_t of VO₂ thin films, with Cr-doped VO₂ and Ge-doped VO₂ showing T_t of ~100 °C and ~95 °C respectively. Besides, T_t may also be modulated via strain engineering. In particular, tensile strain along the c-axis of the VO₂ lattice results in a higher T_t. It has been demonstrated that VO₂ thin films deposited on TiO₂(100)⁶⁰ and TiO₂(110)⁶¹ substrates have increased T_t due to substrate-induced strain, with the latter reporting a T_t of ~95 °C. Such transition temperature range, by means of doping or strain engineering, is more favorable for practical applications.”

3) The paper reports a series of experimental results without in-depth results of the sensing performance and without any basic modeling or discussion of what are the limits of these sensing principle versus existing state of the art sensory systems for light, pressure and temperature. The paper needs to compare and discuss the achieved performance of the proposed VO₂ sensors with state of the art sensor references in terms of achieved sensitivity, selectivity, SNR and power consumption. Without such type of benchmarking and discussion of the real advantages of the proposed sensing solutions the work cannot be published. Also the reference list should be updated with more recent publications of other groups, including spiking VO₂ sensors.

Our response: We would like to greatly thank the reviewer for the constructive comment.

We are sorry for the lack of models in our previous version. To address this

question, we have performed further modeling studies on the spiking neuron. We have added the following equations and discussions in the revised manuscript:

- Page 19-20:

“Moreover, we established a model of the spiking sensory neuron. The artificial spiking neuron circuit is essentially an RC circuit. Using Kirchhoff's Current Law, we have the following differential equation:

$$C_m \frac{dV_m}{dt} = \frac{V_{in} - V_m}{R_L} - \frac{V_m}{R_{VO_2}} \quad (2)$$

where C_m is the capacitance in parallel to the VO_2 device or can be parasitic capacitance. V_m is the output voltage across the VO_2 device. The VO_2 resistance is $R_{VO_2} = R_{off}$ in HRS and $R_{VO_2} = R_{on}$ in LRS. For simplicity, we assume that R_{off} and R_{on} are constant in our analyses.

To obtain the rising time, t_r , from V_{hold} to V_{th} during oscillation, we analyze the circuit when $R_{VO_2} = R_{off}$. By integrating the equation and applying the initial condition $V_m(0) = V_{hold}$, we obtain:

$$V_m(t) = \frac{R_{off}}{R_L + R_{off}} V_{in} - \left(\frac{R_{off}}{R_L + R_{off}} V_{in} - V_{hold} \right) \exp\left(-\frac{t}{(R_L \parallel R_{off})C_m} \right) \quad (3)$$

At $t = t_r$, $V_m(t_r) = V_{th}$. Inserting these values into the equation and further rearranging, we arrive at the expression:

$$t_r = (R_L \parallel R_{off})C_m \cdot \ln\left(\frac{\frac{R_{off}}{R_L + R_{off}} V_{in} - V_{hold}}{\frac{R_{off}}{R_L + R_{off}} V_{in} - V_{th}} \right) \quad (4)$$

For the falling time, t_f , from V_{th} to V_{hold} during oscillation, we let $R_{VO_2} = R_{on}$. By integrating the differential equation at initial condition $V_m(0) = V_{th}$, we arrive at the

following equation:

$$V_m(t) = \frac{R_{on}}{R_L + R_{on}} V_{in} - \left(\frac{R_{on}}{R_L + R_{on}} V_{in} - V_{th} \right) \exp\left(-\frac{t}{(R_L \parallel R_{on})C_m}\right) \quad (5)$$

At $t = t_f$, $V_m(t_f) = V_{hold}$. Hence, we have:

$$t_f = (R_L \parallel R_{on})C_m \cdot \ln\left(\frac{V_{th} - \frac{R_{on}}{R_L + R_{on}} V_{in}}{V_{hold} - \frac{R_{on}}{R_L + R_{on}} V_{in}}\right) \quad (6)$$

Thus, the oscillating frequency is:

$$f = \frac{1}{t_r + t_f} \quad (7)$$

This model is similar to the one given in Ref.54,55. Extending the model to calibratable spiking sensory neuron, we have:

$$R_L = R_{sensor} \parallel R_{scaling} + R_c \quad (8)$$

where R_{sensor} , $R_{scaling}$, and R_c are resistance of sensor, scaling resistance and calibration resistance.”

- Page 22:

“After considering the effect of temperature, the threshold and holding voltages are corrected and can be described as⁶²:

$$V_{th}(T) = \sqrt{\frac{R_{off}}{R_{th}} (T_t - T)} \quad (9)$$

$$V_{hold}(T) = \sqrt{\frac{R_{on}}{R_{th}} (T_t - T)} \quad (10)$$

where R_{th} , T_t , and T are the effective thermal resistance, the transition temperature of VO_2 and the operating temperature respectively. Therefore, the impact of temperature on VO_2 neuron spiking can be obtained by inserting Equation 9-10 into Equation 2-8.

The validity of this model has been verified in Supplementary Fig. 23, where a set of f - R_L curves under different temperatures are calculated using this model (the values of the parameters are shown in Supplementary Table 2). The calculated results are well consistent with the experimentally measured data (Supplementary Fig. 22l), hence demonstrating the reliability of our model.”

Supplementary Figure 23. Spiking frequency as a function of load resistance (R_L) with different temperatures calculated by the model.

The second part of the question is regarding the sensing performance of our approach. To evaluate this quantitatively, the sensitivity is defined as $S = \Delta f / \Delta x$, where Δf and Δx are the values of change in the neuronal frequency and input, respectively. The spiking sensory neuron is able to achieve high sensitivity of 151.74 kHz/N, 0.13 kHz/Lux and 2.8 kHz/°C in tactile, optical, and temperature perception, respectively. Furthermore, the signal-to-noise ratio (SNR) of the spiking sensory neuron is defined for the first time, which can be described as:

$$SNR = 10 \log \frac{\mu(f)^2}{\sigma(f)^2} \quad (R26)$$

where $\mu(f)$, $\sigma(f)$ are the mean and standard deviations of spiking frequency in every

oscillation cycle, respectively. Statistical analysis on the experimental results shows that the artificial spiking sensory neurons can achieve SNRs of 33.66 dB, 31.90 dB and 29.92 dB in the tactile, optical and temperature sensing using our approach.

To address this issue, we have added the following sentence in Page 18-19 of the manuscript: *“To evaluate the sensing performance, the sensitivity of the spiking sensory neurons is defined as $S = \Delta f / \Delta x$, where Δf and Δx are the values of change in the neuronal frequency and input, respectively. The spiking sensory neuron is able to achieve high sensitivity of 151.74 kHz/N, 0.13 kHz/Lux and 2.8 kHz/ °C in tactile, optical, and temperature perception, respectively. Furthermore, the signal-to-noise ratio (SNR)⁵³ of the spiking sensory neuron is defined for the first time, which can be described as:*

$$SNR = 10 \log \frac{\mu(f)^2}{\sigma(f)^2} \quad (1)$$

where $\mu(f)$, $\sigma(f)$ are the mean and standard deviations of spiking frequency in every oscillation cycle, respectively. Statistical analysis on the experimental results shows that the artificial spiking sensory neurons can achieve SNRs of 33.66 dB, 31.90 dB and 29.92 dB in the tactile, optical and temperature sensing using our approach. These SNR values can originate from the sensor, circuit as well as fluctuations in V_{th} , V_{hold} , R_{off} and R_{on} of the VO_2 device.”

In order to benchmark with existing sensors and justify the advantages of the approach proposed here, we compared our approach with traditional silicon circuits and other spiking sensory neurons. Supplementary Fig. 24 schematically depicts the comparison between neuromorphic perception system based on silicon circuits and our

approach. In traditional silicon-based circuits, in order to sense physical signals a large number of ADCs (analog-to-digital converters) are necessary besides the sensors, which are very costly in area and energy consumption, and when the subsequent information processing is in spike-based neuromorphic computing systems, a large number of additional VSCs (voltage-to-spike converters) will be required^{R12} to realize spike conversion, which also consume a large amount of area and energy, as shown in Supplementary Fig. 24a. In stark contrast, our calibratable sensory neuron (Supplementary Fig. 24b) can directly achieve both sensing and spike conversion with the simple circuit consisting of the sensor, the VO₂ memristor and a few resistors and capacitors, which is much more efficient in area and energy consumption compared to silicon circuits.

In addition, we have also compared our approach with other state-of-the-art spike-based sensory neurons, as detailed in Supplementary Table 3. Our approach has advantages in the following aspects:

- 1) Compared with existing works in the literature, our work achieves multiple perception modalities including pressure, light, temperature and curvature for the first time, which is due to the successful solution of the impedance matching problem between sensors and neurons using our neuron circuit. Otherwise, the neuron can fire only when the resistance range of the sensor is consistent with the resistance range of the neuron. Our neuron circuit has effectively addressed this issue by utilizing the scaling resistance and calibration resistance to adapt the working resistance ranges of different sensors, so that a variety of different modalities of perception are successfully

realized, which is a significant advantage of our approach and not seen in existing studies.

2) We have defined and analyzed the signal-to-noise ratio (SNR) of the spiking sensory neuron, which has not been reported in other state-of-the-art spike-based sensory neurons. The high crystalline quality of epitaxial VO₂ has addressed the fundamental cycle-to-cycle and device-to-device variation issues in sensory neurons, and the resultant excellent uniformity of our devices gives rise to excellent SNRs of 33.66 dB, 31.90 dB and 29.92 dB in tactile, optical and temperature sensing, respectively.

3) The epitaxial VO₂ based sensory neurons in this work achieve high sensitivity in different sensing modes, namely, 151.74 kHz/N, 0.13 kHz/Lux and 2.8 kHz/°C for tactile, optical, and temperature perception, respectively. Such sensitivity can be further improved by using smaller capacitors in parallel, as smaller capacitors can increase the firing frequency in general. The highest firing frequency of our neurons reaches 1.3 MHz, which can be improved by reducing the parallel capacitance as well. Currently, our VO₂ devices have relatively high energy consumption, with each spike consuming ~2.9 nJ, but our additional experimental studies have shown that such energy consumption can be further optimized by i) increasing the device resistance through optimizing the growth conditions of the VO₂ film (Supplementary Fig. 15a-d), and ii) reducing the threshold voltage via decreasing the channel length of the VO₂ memristor (Supplementary Fig. 15a-e).

To address this clearly in the revised manuscript, we have included Supplementary

Fig. 24 and Supplementary Table 3 in Page 27-28, along with the following discussion:

“The calibratable artificial sensory neurons based on epitaxial VO₂ shown here demonstrate significant advantages compared with traditional silicon circuits and other spiking sensory neurons. Supplementary Fig. 24 schematically depicts the comparison between neuromorphic perception system based on silicon circuits and our approach. In traditional silicon-based circuits, in order to sense physical signals a large number of ADCs (analog-to-digital converters) are necessary besides the sensors, which are very costly in area and energy consumption, and when the subsequent information processing is in spike-based neuromorphic computing systems, a large number of additional VSCs (voltage-to-spike converters) will be required³⁰ to realize spike conversion, which also consume a large amount of area and energy, as shown in Supplementary Fig. 24a. In stark contrast, our calibratable sensory neuron (Supplementary Fig. 24b) can directly achieve both sensing and spike conversion with the simple circuit consisting of the sensor, the VO₂ memristor and a few resistors and capacitors, which is much more efficient in area and energy consumption compared to silicon circuits. Supplementary Table 3 further benchmarks our approach with other state-of-the-art spike-based sensory neurons. Compared with existing works in the literature, our work effectively addressed the impedance matching problem between sensors and neurons by utilizing the scaling resistance and calibration resistance to adapt the working resistance ranges of different sensors. As a result, a variety of different perception modalities including pressure, light, temperature and curvature have been achieved for the first time, which is a significant advantage of our approach

and not seen in existing studies. The high crystalline quality of epitaxial VO_2 has addressed the fundamental cycle-to-cycle and device-to-device variation issues in sensory neurons, and the resultant excellent uniformity of our devices gives rise to excellent SNRs of 33.66 dB, 31.90 dB and 29.92 dB in tactile, optical and temperature sensing, respectively. Our investigations have revealed that the present energy consumption, sensitivity and firing frequency of the sensory neurons could be further improved by optimizing the growth conditions of the VO_2 film (Supplementary Fig. 15a-d), the channel length (Supplementary Fig. 15a-e), and circuit parameters such as parallel capacitance (Fig. 3, Supplementary Fig. 7-11).”

Supplementary Figure 24. Comparison between neuromorphic perception system based on silicon circuits and our system. (a) Schematic of the traditional neuromorphic perception system based on silicon circuits with lots of ADCs (analog-

to-digital converters) and VSCs (voltage-to-spike converters). **(b)** Schematic of neuromorphic perception system based on our calibratable artificial sensory neurons.

Supplementary Table 3. Comparison of different spike-based sensory neurons

Device		Epitaxial VO ₂ (this work)	VO ₂ ¹	NbO _x ²	NbO _x ³	NbO _x ⁴
Perception type	Pressure	✓	✓	✓	×	✓
	Light	✓	×	×	✓	×
	Temperature	✓	×	×	×	×
	Curvature	✓	×	×	×	×
Impedance matching with different sensors		✓	×	×	×	×
Sensitivity		151.74 kHz/N, 0.13 kHz/Lux, 2.8 kHz/°C	60.8 kHz/kPa	/	/	/
SNR		33.66 dB (tactile), 31.90 dB (optical), 29.92 dB (temperature)	/	/	/	/
Energy consumption		2.9 nJ/spike	/	38 pJ/spike	/	~6 nJ/ spike
Highest spiking frequency		~1.3 MHz	174 kHz	~1.1 MHz	~2.2 MHz	~9 MHz

The reference in the form is as follows:

1. Fang, S.L. *et al.* An Artificial Spiking Afferent Neuron System Achieved by 1M1S for Neuromorphic Computing. *IEEE Trans. Electron. Devices*. 1-7 (2022).

2. Zhang, X. *et al.* An artificial spiking afferent nerve based on Mott memristors for neurorobotics. *Nat. Commun.* **11**, 51 (2020).
3. Wu, Q. *et al.* Spike Encoding with Optic Sensory Neurons Enable a Pulse Coupled Neural Network for Ultraviolet Image Segmentation. *Nano. Lett.* **20**, 8015-8023 (2020).
4. Li, F. *et al.* A Skin-Inspired Artificial Mechanoreceptor for Tactile Enhancement and Integration. *ACS Nano.* **15**, 16422-16431 (2021).

Lastly, we also greatly appreciate the reviewer's suggestion on updating the references. We have included the following latest papers into the reference list and the number of the references has been updated accordingly in the revised manuscript.

30. Fang, S.L. *et al.* An Artificial Spiking Afferent Neuron System Achieved by 1M1S for Neuromorphic Computing. *IEEE Trans Electron Devices*, 1-7 (2022).

31. Li, F. *et al.* A Skin-Inspired Artificial Mechanoreceptor for Tactile Enhancement and Integration. *ACS Nano.* **15**, 16422-16431 (2021).

Comments from Reviewer #2

Overall Remarks: The manuscript by R. Yuan reports on capabilities of neuromorphic sensory system on reproducing biological functions incorporating different excitation stimuli such as temperature, light and pressure. The sensory neurons were based on VO₂ epitaxial films prepared by PLD. The shown applications are nicely demonstrating the capability of the system for emulating more complex neural functionalities. The manuscript is well structured and written, the topic is in suitable for the journal. I would recommend publication provided the comments below are appropriately addressed.

Our response: we would like to sincerely thank the reviewer for the positive assessment. We also deeply appreciate the valuable comments this reviewer made and performed new experiments, modeling as well as simulations (Supplementary Fig. 21-23, Supplementary Table 2, Equation 2-10). Our responses to the comments one by one are shown as follows.

Technical Comments

1. VO₂ is a system that is very sensitive to both temperature and oxygen content. Deviations in these parameters may lead to a change in the crystal structure and/or change in the stoichiometry. Both factors lead to a change in the electronic and defect-chemical structure. Did the authors considered this specifics of VO₂?

Our response: We would like to greatly thank the reviewer for the constructive comment. Indeed, VO₂ is very sensitive to both temperature and oxygen content. In order to examine these two factors, we have performed new experiments to measure

the characteristics of VO₂ devices under different temperatures and atmospheric pressure. The new experimental data are now included in Supplementary Fig. 21-22. Supplementary Fig. 21a shows the I - V characteristics of the VO₂ memristor in air, while Supplementary Fig. 21b-h shows the I - V characteristics at atmospheric pressure from 1.5×10^{-3} mbar to 2×10^{-4} mbar, so that the oxygen content is gradually reduced. The VO₂ device showed no significant change in its I - V characteristics. To further explore the effect of oxygen content on the device, we further measured the device characteristics in N₂ environment so that oxygen is avoided (Supplementary Fig. 21i), where similar I - V characteristics were observed once again. To quantify the impact, the positive/negative threshold and holding voltages (V_{th_pos} , V_{th_neg} , V_{hold_pos} and V_{hold_neg}) as well as R_{off} and R_{on} of the devices at different atmospheric pressures are extracted and displayed in Supplementary Fig. 21j-k. The highly stable threshold and holding voltages as well as resistance states demonstrate that the VO₂ memristor can operate stably under varied oxygen content.

The second part of the question is regarding the impact of temperature on VO₂ memristor. In light of this advice, we have performed new experiments and modeling studies (Supplementary Fig. 22-23). One can see from Supplementary Fig. 22a-h that the voltage window of V_{th} - V_{hold} shrinks gradually as temperature increases from 283 to 305 K, which can be more evidently seen in Supplementary Fig. 22i. Both V_{th} and V_{hold} gradually decrease with increased temperature. Moreover, the firing frequency of the VO₂ spiking neuron at different temperatures is further tested from 283 to 291 K with constant R_L of 4 k Ω and the same input voltage of 5 V (Supplementary Fig. 22k), where

the devices are placed directly on a temperature-controlled probe station (Supplementary Fig. 22j). As the temperature increases, one can find that the firing frequency of VO₂ neurons gradually increases (576.13–656.02 kHz). We have systematically tested the dependence of the spiking frequency as a function of load resistance (R_L) and temperature, and the results are displayed in Supplementary Fig. 22l, showing similar R_L and T dependence in all cases. This might be ascribed to the gradual decrease in threshold voltages of VO₂ memristors with increased temperature, so that the neuronal circuit requires lower voltage to fire.

In order to understand such temperature dependence, we have further constructed a model on the VO₂ spiking neuron. The artificial spiking neuron circuit is essentially an RC circuit. Using Kirchhoff's Current Law, we have the following differential equation:

$$C_m \frac{dV_m}{dt} = \frac{V_{in} - V_m}{R_L} - \frac{V_m}{R_{VO_2}} \quad (R1)$$

where C_m is the capacitance in parallel to the VO₂ device or can be a parasitic capacitance. V_m is the output voltage across the VO₂ device. The VO₂ resistance is $R_{VO_2} = R_{off}$ in HRS and $R_{VO_2} = R_{on}$ in LRS. For simplicity, we assume that R_{off} and R_{on} are constant in our analyses.

To obtain the rising time, t_r , from V_{hold} to V_{th} during oscillation, we analyze the circuit when $R_{VO_2} = R_{off}$. By integrating the equation and applying the initial condition $V_m(0) = V_{hold}$, we obtain:

$$V_m(t) = \frac{R_{off}}{R_L + R_{off}} V_{in} - \left(\frac{R_{off}}{R_L + R_{off}} V_{in} - V_{hold} \right) \exp\left(-\frac{t}{(R_L \parallel R_{off})C_m} \right) \quad (R2)$$

At $t = t_r$, $V_m(t_r) = V_{th}$. Inserting these values into the equation and further rearranging, we arrive at the expression:

$$t_r = (R_L \parallel R_{off})C_m \cdot \ln \left(\frac{\frac{R_{off}}{R_L + R_{off}} V_{in} - V_{hold}}{\frac{R_{off}}{R_L + R_{off}} V_{in} - V_{th}} \right) \quad (R3)$$

For the falling time, t_f , from V_{th} to V_{hold} during oscillation, we let $R_{VO_2} = R_{on}$. By integrating the differential equation at initial condition $V_m(0) = V_{th}$, we arrive at the following equation:

$$V_m(t) = \frac{R_{on}}{R_L + R_{on}} V_{in} - \left(\frac{R_{on}}{R_L + R_{on}} V_{in} - V_{th} \right) \exp \left(-\frac{t}{(R_L \parallel R_{on})C_m} \right) \quad (R4)$$

At $t = t_f$, $V_m(t_f) = V_{hold}$. Hence, we have:

$$t_f = (R_L \parallel R_{on})C_m \cdot \ln \left(\frac{V_{th} - \frac{R_{on}}{R_L + R_{on}} V_{in}}{V_{hold} - \frac{R_{on}}{R_L + R_{on}} V_{in}} \right) \quad (R5)$$

Thus, we can calculate the oscillating frequency:

$$f = \frac{1}{t_r + t_f} \quad (R6)$$

This model is similar to the one given in Ref. R1, R2. Extending the model to calibratable spiking sensory neuron, we have:

$$R_L = R_{sensor} \parallel R_{scaling} + R_c \quad (R7)$$

where R_{sensor} , $R_{scaling}$, and R_c are resistance of sensor, scaling resistance and calibration resistance.

Since V_{th} and V_{hold} are temperature-dependent parameters (Supplementary Fig. 22i), the threshold/holding voltages are corrected and can be described as^{R4}:

$$V_{th}(T) = \sqrt{\frac{R_{off}}{R_{th}}(T_t - T)} \quad (R9)$$

$$V_{hold}(T) = \sqrt{\frac{R_{on}}{R_{th}}(T_t - T)} \quad (R10)$$

These relations were derived from the heat equation using lumped analysis where R_{th} , T_t , and T are the effective thermal resistance, the transition temperature of VO₂ and the operating temperature, respectively. Therefore, the complete spiking sensory neuron model after considering the effect of temperature can be obtained by inserting Equation R9-R10 into Equation R1-R7.

The validity of this model has been verified in Supplementary Fig. 23, where a set of f - R_L curves under different temperatures are calculated using this model (the values of the parameters are shown in Supplementary Table 2). The calculated results are well consistent with the experimentally measured data (Supplementary Fig. 22), hence demonstrating the reliability of our model. This can therefore be used to model and assess the impact of temperature in sensing applications.

We have added the new experimental and modeling results in Supplementary Fig. 21-23 and Supplementary Table 2, and we also added the following discussion into the revised manuscript:

- Page 19-20:

“Moreover, we established a model of spiking sensory neuron. The artificial spiking neuron circuit is essentially an RC circuit. Using Kirchhoff's Current Law, we have the following differential equation:

$$C_m \frac{dV_m}{dt} = \frac{V_{in} - V_m}{R_L} - \frac{V_m}{R_{VO_2}} \quad (2)$$

where C_m is the capacitance in parallel to the VO_2 device and can be parasitic capacitance. V_m is the output voltage across the VO_2 device. The VO_2 resistance is $R_{VO_2} = R_{off}$ in HRS and $R_{VO_2} = R_{on}$ in LRS. In reality, R_{off} and R_{on} is dependent on temperature, which is in turn dependent on current. Hence, these values dynamically evolve depending on the state of the circuit. For simplicity, we assume that these values are constant in our analyses.

To obtain the rising time, t_r , from V_{hold} to V_{th} during oscillation, we analyze the circuit when $R_{VO_2} = R_{off}$. By integrating the equation and applying the initial condition $V_m(0) = V_{hold}$, we obtain:

$$V_m(t) = \frac{R_{off}}{R_L + R_{off}} V_{in} - \left(\frac{R_{off}}{R_L + R_{off}} V_{in} - V_{hold} \right) \exp\left(-\frac{t}{(R_L \parallel R_{off})C_m}\right) \quad (3)$$

At $t = t_r$, $V_m(t_r) = V_{th}$. Plugging these values into the equation and further rearranging, we arrive at the expression:

$$t_r = (R_L \parallel R_{off})C_m \cdot \ln\left(\frac{\frac{R_{off}}{R_L + R_{off}} V_{in} - V_{hold}}{\frac{R_{off}}{R_L + R_{off}} V_{in} - V_{th}}\right) \quad (4)$$

For the falling time, t_f , from V_{th} to V_{hold} during oscillation, we let $R_{VO_2} = R_{on}$. By integrating the differential equation at initial condition $V_m(0) = V_{th}$, we arrive at the following equation:

$$V_m(t) = \frac{R_{on}}{R_L + R_{on}} V_{in} - \left(\frac{R_{on}}{R_L + R_{on}} V_{in} - V_{th} \right) \exp\left(-\frac{t}{(R_L \parallel R_{on})C_m}\right) \quad (5)$$

At $t = t_f$, $V_m(t_f) = V_{hold}$. Hence, we have:

$$t_f = (R_L \parallel R_{on})C_m \cdot \ln\left(\frac{V_{th} - \frac{R_{on}}{R_L + R_{on}} V_{in}}{V_{hold} - \frac{R_{on}}{R_L + R_{on}} V_{in}}\right) \quad (6)$$

Thus, oscillating frequency could be:

$$f = \frac{1}{t_r + t_f} \quad (7)$$

This model is similar to the one given in Ref.54,55. Extending to calibratable spiking sensory neuron, we have:

$$R_L = R_{\text{sensor}} || R_{\text{scaling}} + R_c \quad (8)$$

where R_{sensor} , R_{scaling} , and R_c are resistance of sensor, scaling resistance and calibration resistance.”

- Page 20:

“It is worth noting that VO₂ is a system that is very sensitive to oxygen content, protons and temperature in ambient environment. In order to examine these factors, we have firstly performed control experiments to measure the characteristics of VO₂ devices under different atmospheric pressure, including air (Supplementary Fig. 21a), varied atmospheric pressure from 1.5×10^{-3} mbar to 2×10^{-4} mbar (Supplementary Fig. 21b-h) and N₂ environment (Supplementary Fig. 21i). Therefore, the concentration of oxygen and moisture/proton is gradually reduced in this process, where the VO₂ device showed no significant change in its I-V characteristics. To quantify the impact, the threshold and holding voltages ($V_{\text{th_pos}}$, $V_{\text{th_neg}}$, $V_{\text{hold_pos}}$ and $V_{\text{hold_neg}}$) as well as R_{off} and R_{on} of the devices at different atmospheric pressures are extracted (Supplementary Fig. 21j-k). The highly stable threshold and holding voltages as well as resistance states demonstrate that the VO₂ memristor can operate stably under varied oxygen and moisture concentrations.”

- Page 21:

“To model and address the effect of temperature, we tested the I-V characteristics of the device at different temperatures. The I-V characteristics of VO₂ memristor at 283-305 K are displayed in Supplementary Fig. 22a-h. Supplementary Fig. 22i shows V_{th} and V_{hold} at different temperatures, where one can find that both V_{th} and V_{hold} gradually decrease with increased temperature. Moreover, the firing frequency of the VO₂ spiking neuron at different temperatures is further tested from 283 to 291 K with constant R_L of 4 kΩ and the same input voltage of 5 V (Supplementary Fig. 22k), where the devices are placed directly on a temperature-controlled probe station (Supplementary Fig. 22j). As the temperature increases, one can find that the firing frequency of VO₂ neurons gradually increases (576.13–656.02 kHz). We have systematically tested the dependence of the spiking frequency as a function of load resistance (R_L) and temperature, and the results are displayed in Supplementary Fig. 22l, showing similar R_L and T dependence in all cases. This might be ascribed to the gradual decrease in threshold voltages of VO₂ memristors with increased temperature, so that the neuronal circuit requires lower voltage to fire.”

- Page 22:

“After considering the effect of temperature, the threshold and holding voltages are corrected and can be described as⁶²:

$$V_{th}(T) = \sqrt{\frac{R_{off}}{R_{th}}(T_t - T)} \quad (9)$$

$$V_{\text{hold}}(T) = \sqrt{\frac{R_{\text{on}}}{R_{\text{th}}}(T_t - T)} \quad (10)$$

where R_{th} , T_t , and T are the effective thermal resistance, the transition temperature of VO_2 and the operating temperature respectively. Therefore, the impact of temperature on VO_2 neuron spiking can be obtained by inserting Equation 9-10 into Equation 2-8. The validity of this model has been verified in Supplementary Fig. 23, where a set of f - R_L curves under different temperatures are calculated using this model (the values of the parameters are shown in Supplementary Table 2). The calculated results are well consistent with the experimentally measured data (Supplementary Fig. 22l), hence demonstrating the reliability of our model.”

Supplementary Figure 21. Characteristics of the VO₂ memristor under different environment. (a) The I - V characteristics of devices in normal atmospheric pressure. **(b)-(i)** The I - V characteristics of devices in different atmospheric pressure which is changed from 1.5×10^{-3} mbar to 2×10^{-4} and N_2 environment. The I - V characteristics of devices are stable. **(j)** The V_{th_pos} , V_{th_neg} , V_{hold_pos} and V_{hold_neg} at different atmospheric pressures. **(k)** The resistive states of devices at different atmospheric pressures.

Supplementary Figure 22. The characteristics of the VO₂ memristor under different temperatures. (a)-(h) The I - V characteristics of devices with different temperature. The window of V_{th} - V_{hold} shrinks gradually with increasing temperature. **(i)** The V_{th} and V_{hold} at different temperatures. Both V_{th} and V_{hold} gradually decreases with increasing temperature. **(j)** Diagram of equipment to test firing frequency of spiking neuron at different temperatures. Devices are placed directly on a temperature-

controlled platform. **(k)** The firing frequency of neurons at different temperatures. With increasing temperature, the firing frequency gradually increased where input voltage is 5 V and R_L is 4 k Ω . **(i)** Spiking frequency as a function of load resistance (R_L) with different temperatures.

Supplementary Figure 23. Spiking frequency as a function of load resistance (R_L) with different temperatures calculated by the model.

Supplementary Table 2. The values of the parameters used in the modeling of spiking neuron considering the temperature.

Parameter	Value	Units
R_{off}	3100	Ω
R_{on}	450	Ω
R_{th}	31313	K/W
T_i	318.736	K
V_{in}	5	V
R_L	4	k Ω

2. In same context – properties of VO₂ can also be changed by local environment and/or protons. Was this factor considered by the experiments?

Our response: We would like to thank the reviewer for the very constructive suggestion. Indeed, the local environment and/or protons is another factor(s) that can affect VO₂ properties. In light of this, we have performed new studies to measure the characteristics of VO₂ devices under different atmospheric pressure, including air (Supplementary Fig. 21a), varied atmospheric pressure from 1.5×10^{-3} mbar to 2×10^{-4} mbar (Supplementary Fig. 21b-h) and N₂ environment (Supplementary Fig. 21i), and hence the local environment and/or concentration of moisture/proton is systematically modulated. The experimental results showed that the VO₂ device did not exhibit significant change in electrical characteristics. To quantify the impact, the threshold and holding voltages (V_{th_pos} , V_{th_neg} , V_{hold_pos} and V_{hold_neg}) as well as R_{off} and R_{on} of the devices at different atmospheric pressures are extracted (Supplementary Fig. 21j-k), once again demonstrating that the VO₂ memristor can operate stably under varied moisture/proton concentrations.

To address this question, the new results are now included as Supplementary Fig. 21 in the revised manuscript, along with the following discussion in Page 20 of the main text: *“It is worth noting that VO₂ is a system that is very sensitive to oxygen content, protons and temperature in ambient environment. In order to examine these factors, we have firstly performed control experiments to measure the characteristics of VO₂ devices under different atmospheric pressure, including air (Supplementary Fig. 21a), varied atmospheric pressure from 1.5×10^{-3} mbar to 2×10^{-4} mbar (Supplementary Fig.*

21b-h) and N_2 environment (Supplementary Fig. 21i). Therefore, the concentration of oxygen and moisture/proton is gradually reduced in this process, where the VO_2 device showed no significant change in its I-V characteristics. To quantify the impact, the threshold and holding voltages (V_{th_pos} , V_{th_neg} , V_{hold_pos} and V_{hold_neg}) as well as R_{off} and R_{on} of the devices at different atmospheric pressures are extracted (Supplementary Fig. 21j-k). The highly stable threshold and holding voltages as well as resistance states demonstrate that the VO_2 memristor can operate stably under varied oxygen and moisture concentrations.”

Supplementary Figure 21. Characteristics of the VO₂ memristor under different environment. (a) The I - V characteristics of devices in normal atmospheric pressure. **(b)-(i)** The I - V characteristics of devices in different atmospheric pressure which is changed from 1.5×10^{-3} mbar to 2×10^{-4} and N_2 environment. The I - V characteristics of devices are stable. **(j)** The V_{th_pos} , V_{th_neg} , V_{hold_pos} and V_{hold_neg} at different atmospheric pressures. **(k)** The resistive states of devices at different atmospheric pressures.

3. The authors are pointing out that VO₂ is showing metal-insulator transition.

Within the field of RRAMS it is known that Mott-transition is meant but the more general audience of Nature Communications would be probably confused. I would recommend adding a short paragraph explaining the physics of the resistance change in VO₂.

Our response: We would like to greatly thank the reviewer for the valuable suggestion.

To strengthen the discussion on the physics of resistance change in VO₂ in our manuscript, we have added thermodynamic simulation by COMSOL (Supplementary Fig. 2), along with further discussions on the physical mechanism in Page 7-8 of the main text: *“Such volatile threshold switching (TS) characteristics and metal-insulator transition in VO₂ have attracted extensive attention^{40, 41}, which has a complex mechanism involving both electronic and structural phase transitions⁴². Supplementary Fig. 2 shows the experimental results and simulated I-V curve based on the metal-insulator transition (MIT) model, where the blue points are the experimental data and the red curve is the simulation result, along with spatial heat distribution in different stages of the phase transition. As the applied voltage progressively increases (state (1) to (2)), heat is generated in the VO₂ memristor. Once the phase transition is triggered, a filament is formed through the VO₂ gap, which switches the device from HRS to LRS. The filament is expanded as the voltage increases (state (2) to state (3)). When the applied voltage is reduced, the heat dissipates and the filament size decreases (state (3) to state (4)). When the applied voltage is below V_{hold} , the filament breaks down and the device eventually returns to HRS (state (4) to state (1)), as shown in Supplementary Fig.*

2. More details of the model used for simulation are provided in Methods, Supplementary Table 1, and Supplementary Note 1.”

Supplementary Figure 2. Simulation results. The experimental I - V curve can be well fitted using our model. The insets show spatial heat distribution in different stages of the phase transition.

4. PLD is a very suitable method for preparing high quality films in lab conditions.

However, it is limited for use for applications as it cannot cover homogeneously larger surfaces (in the general case it can reliably cover about $2 \times 2 \text{ cm}^2$ substrate).

It would be good to discuss what other methods can be used for larger samples.

Our response: We would like to greatly thank the reviewer for the valuable suggestion.

Although many methods have been adopted to synthesize high-quality VO_2 films, the growth of wafer-scale, high-quality VO_2 films with excellent phase transition property is still a challenge. The molecular beam epitaxy (MBE) method is a widely used technique to produce high-quality and homogeneous epitaxial films. 2-inch epitaxial VO_2 film using an RF-plasma assisted oxide MBE method has been prepared by Fan et al^{R13}. There are also other film growth methods for preparing large-scale VO_2 , such as

electron-beam evaporation, thermal oxidation, sol-gel method and sputtering etc. For example, They et al. reported growth of large-area crystalline, uniform VO₂ layers on different types of substrates by electron-beam evaporation^{R14}. Besides, a simple method for producing uniform VO₂ films on 2-inch wafers using water vapor assisted thermal oxidation was proposed by Ren et al^{R15}, where the oxidizing agent used was water vapor and it effectively oxidized the metallic V film to VO₂ film. The sol-gel method is another method for growing large-area VO₂ thin films with low cost and simple preparation process. Wu et al. prepared M1 phase VO₂ thin films on Al₂O₃ substrates using simple sol-gel method^{R16}. In addition, sputtering has been used to grow VO₂ thin films since 1967^{R17} and has shown capability in preparing VO₂ films on 4-inch wafers^{R18}. Nevertheless, the crystalline quality of the VO₂ film might be compromised in some of the above preparation processes, and the growth method should be selected based on the detailed requirements on sample scale and crystalline quality in the applications.

To address this clearly, we have added the following sentences into Page 9 of the revised manuscript: *“It should be noted that despite the high film quality, PLD is still limited in preparing large-scale thin films. Many methods have been adopted to synthesize high-quality VO₂ films, however, the growth of wafer-scale, high-quality VO₂ films with excellent phase transition property is still a challenge. To date, 2-inch epitaxial VO₂ film grown by molecular beam epitaxy was reported⁴⁴, and preparation of large-scale VO₂ films by electron-beam evaporation⁴⁵, thermal oxidation⁴⁶, sol-gel method⁴⁷ and sputtering⁴⁸ has also been reported. Nevertheless, the crystalline quality*

of the VO₂ film might be compromised in some preparation processes, and the growth method should be selected based on the detailed requirements on sample scale and crystalline quality in the applications.”

5. From the title and the first part of the article I have had the impression that VO₂ is used as both neuron and sensor in same time. Later in the article it becomes clear that different sensors are used in a combination with the VO₂ memristor. I would recommend presenting this at earlier stage.

Our response: We are sorry for the ambiguity this has caused. To avoid misunderstanding, we have added the following sentence into the introduction of Page 4: *“In this study, we report a calibratable artificial sensory neuron (CASN) consisting of epitaxial VO₂ memristor grown by pulsed laser deposition and a variety of coupled sensors.”*

6. On page 8 is written: “...variation, which can be attributed to the high crystalline quality of the epitaxial VO₂...”

It would be good to provide a short argumentation (or citation) how the good crystallinity is related to the low C2C variation.

Our response: We would like to sincerely thank the reviewer for the detailed suggestions. To address this question, we have added the following discussion in page 8 of revised manuscript: *“Since the resistance switching in VO₂ is ascribed to the intrinsic electronic and structural phase transitions⁴² in the material itself without*

necessarily incorporating defects unlike redox based memristors, the low C2C variation can be attributed to the high crystalline quality of the epitaxial VO₂.”

Minor issue

- On page 5 is written: “...physical signals into electrical signals...”

The expression is not precise. Electricity is also a physical phenomenon. Therefore an electrical signals is a kind of physical signal.

Our response: We would like to sincerely thank the reviewer for the valuable suggestion. We have revised the expression of the article in page 5 as “*In the biological perception system, certain types of receptors (photoreceptors, thermal receptors, mechanoreceptors, etc.) and neurons convert external environmental signals into electrical spikes (Fig. 1a).*”

Comments from Reviewer #3

Remarks: **This article analyses a very timely and important topic. The model proposed for the artificial spiking sensory neuron seems to be coherent, and the functioning of the VO₂ device is critical for the correct functioning of the whole circuit and neural network.**

The devices have a planar structure in the form of a cap. While this is different from most devices with vertical structure, the lateral size of the devices is still enough small to ensure a reasonably high integration density. In fact, for me it is very surprising that the authors achieve such a low variability in planar devices. That is very meritorious and in my opinion it is related to the high crystallinity of the VO₂ film. I may recommend the authors to quantify the variability by calculating the coefficient of variance of the V_{SET} and V_{RESET}, indicating the number of devices and cycles per device, and compare the obtained results with the available literature, as done in Chen et al. Nature Electronics 638–645 (2020). Also, the controllability of the frequency of the spikes generated is very good, the authors demonstrate to have full control, and the reproducibility seems to be excellent. The experiments with temperature, light and pressure are an added value potentiating the application of these systems in different applications.

The changes here mentioned are optional, as the manuscript is really good. It is really difficult to achieve such low variability in planar devices. I believe a large amount of people would be interested on this paper. Hence, I think Nature Communications would do well publishing it.

Our response: We would like to sincerely thank the reviewer for the positive evaluation of our study. We deeply appreciate the valuable comments this reviewer made in terms of quantifying the coefficient of variance and comparing the results with prior arts in the literature. In light of this valuable advice, we have calculated the coefficient of variation for four different threshold/holding voltages, i.e. V_{th_pos} , V_{hold_pos} , V_{th_neg} and V_{hold_neg} , in 1000 cycles and 10 different devices. The new statistical results are included as Supplementary Figs. 4 and 5. Following the protocol introduced in Ref. R19, we calculated the coefficient of variation (Cv) as the standard deviation (σ) divided by the mean value (μ). The minimum cycle-to-cycle variability in V_{th_pos} , V_{th_neg} , V_{hold_pos} and V_{hold_neg} was 0.73%, 0.7%, 0.51% and 0.5%, respectively, demonstrating very low variability (Supplementary Fig. 4). Supplementary Fig. 5 shows the I - V characteristics and statistical analysis of variability in the positive and negative threshold/holding voltages of 10 epitaxial VO₂ memristors. The device-to-device variability in V_{th_pos} , V_{th_neg} , V_{hold_pos} and V_{hold_neg} was 5.32%, 5.12%, 6.96% and 7.16%, respectively. Notably, Chen et al. has reported low C2C variability of 1.53% and low D2D variability of 5.74% in hexagonal boron nitride based crossbar arrays (Ref. R19). Our present epitaxial VO₂ based memristor hence demonstrates extremely low C2C variability due to its high crystalline structure and reasonably low D2D variability.

To address this clearly in the revised manuscript , the new results are now included as Supplementary Figs. 4-5, along with the following discussion in Page 9 of the main text: *“Following the protocol introduced in previous studies⁴³, we calculated the coefficient of variation (Cv) as the standard deviation (σ) divided by the mean value (μ).*

The minimum cycle-to-cycle variability in V_{th_pos} , V_{th_neg} , V_{hold_pos} and V_{hold_neg} was 0.73%, 0.7%, 0.51% and 0.5%, respectively, demonstrating very low variability (Supplementary Fig. 4). The device-to-device variability in V_{th_pos} , V_{th_neg} , V_{hold_pos} and V_{hold_neg} was 5.32%, 5.12%, 6.96% and 7.16%, respectively (Supplementary Fig. 5). Notably, Chen et al. has reported low C2C variability of 1.53% and low D2D variability of 5.74% in hexagonal boron nitride based crossbar arrays⁴³. Our present epitaxial VO₂ based memristor hence demonstrates extremely low C2C variability and reasonably low D2D variability due to its high crystalline structure.”

Supplementary Figure 4. The statistical analysis of variability in positive and negative threshold/holding voltages in 1000 cycles. (a) The variability of V_{th_pos} . (b) The variability of V_{th_neg} . (c) The variability of V_{hold_pos} , (d) The variability of V_{hold_neg} .

Supplementary Figure 5. *I-V* characteristics and statistical analysis of variability in threshold/holding voltages of 10 epitaxial VO₂ memristors. (a) *I-V* characteristics of the device measured in 10 different epitaxial VO₂ devices. (b) The V_{th_pos} , V_{hold_pos} , V_{th_neg} and V_{hold_neg} in 10 epitaxial VO₂ devices. (c)-(f) The statistical analysis of variability of V_{th_pos} , V_{th_neg} , V_{hold_pos} , and V_{hold_neg} in 10 epitaxial VO₂ devices.

References

- R1. Chen, P.-Y., Seo, J.-S., Cao, Y. & Yu, S. Compact oscillation neuron exploiting metal-insulator-transition for neuromorphic computing. In: *Proceedings of the 35th International Conference on Computer-Aided Design (ICCAD)* (2016).
- R2. Gao, L., Chen, P.-Y. & Yu, S. NbOx based oscillation neuron for neuromorphic computing. *Appl. Phys. Lett.* **111**, 103503 (2017).
- R3. Box, G. Signal-to-Noise Ratios, Performance Criteria, and Transformations. *Technometrics.* **30**, 1-17 (1988).
- R4. Lee, S. B., Kim, K., Oh, J. S., Kahng, B. & Lee, J. S. Origin of variation in switching voltages in threshold-switching phenomena of VO₂ thin films. *Appl. Phys. Lett.* **102**, 063501 (2013).
- R5. Miyazaki, K., Shibuya, K., Suzuki, M., Wado, H. & Sawa, A. Correlation between thermal hysteresis width and broadening of metal-insulator transition in Cr- and Nb-doped VO₂ films. *Jpn. J. Appl. Phys.* **53**, 071102 (2014).
- R6. Brown, B. L., *et al.* Electrical and optical characterization of the metal-insulator transition temperature in Cr-doped VO₂ thin films. *J. Appl. Phys.* **113**, 173704 (2013).
- R7. Krammer, A., *et al.* Elevated transition temperature in Ge doped VO₂ thin films. *J. Appl. Phys.* **122**, 045304 (2017).
- R8. Du, J., *et al.* Significant changes in phase-transition hysteresis for Ti-doped VO₂ films prepared by polymer-assisted deposition. *Sol. Energ. Mat. Sol. C.* **95**, 469-475 (2011).
- R9. Quackenbush, N. F., *et al.* Stability of the M2 phase of vanadium dioxide induced by coherent epitaxial strain. *Phys. Rev. B.* **94**, 085105 (2016).
- R10. Muraoka, Y., Hiroi, Z. Metal-insulator transition of VO₂ thin films grown on TiO₂ (001) and (110) substrates. *Appl. Phys. Lett.* **80**, 583-585 (2002).
- R11. Muraoka, Y. *et al.* Preparation of TaO₂ thin films using NbO₂ template layers by a pulsed laser deposition technique. *Thin Solid Films.* **599**, 125-132 (2016).
- R12. Fang, S.L. *et al.* An Artificial Spiking Afferent Neuron System Achieved by 1M1S for Neuromorphic Computing. *IEEE Trans. Electron. Devices.* **69**, 1-7 (2022).
- R13. Fan, L.L. *et al.* Growth and phase transition characteristics of pure M-phase

- VO₂ epitaxial film prepared by oxide molecular beam epitaxy. *Appl. Phys. Lett.* **103**, 131914 (2013).
- R14. Théry, V. *et al.* Structural and electrical properties of large area epitaxial VO₂ films grown by electron beam evaporation. *J. Appl. Phys.* **121**, 055303 (2017).
- R15. Ren, H. *et al.* Wafer-size VO₂ film prepared by water-vapor oxidant. *Appl. Surf. Sci.* **525**, 146642 (2020).
- R16. Wu, Y.F., Fan, L.L., Chen, S.M., Chen, S., Zou, C.W. & Wu, Z.Y. Spectroscopic analysis of phase constitution of high quality VO₂ thin film prepared by facile sol-gel method. *AIP Advances*. **3**, 042132 (2013).
- R17. Fuls, E. N., Hensler, D.H. & Ross, A.R. Reactively Sputtered Vanadium Dioxide Thin Films. *Appl Phys Lett.* **10**, 199-201 (1967).
- R18. Cao, C. *et al.* Simple and Low-temperature Growth of High Thermal Performance W-Ti Co-Doped VO₂ Films on a Scale Wafer. *Nano. Advances*. **2**, 23-28 (2017).
- R19. Chen, S. *et al.* Wafer-scale integration of two-dimensional materials in high-density memristive crossbar arrays for artificial neural networks. *Nat. Electron.* **3**, 638-645 (2020).

REVIEWERS' COMMENTS

Reviewer #1 (Remarks to the Author):

The authors have addressed in details many of the important raised questions and the paper is improved, this is very appreciated.

1) I appreciate that the power limit is discussed now better in the paper. On the other hand, it appears that the power per spike is not yet very competitive for neuromorphic applications and even matching the state of the art (few nanoJoule reported is far from neuromorphic literature showing few pJ/spike as state of the art) - However, the discussion provided and the rationale how to scale are now improved. Please try to carefully reflect this gap in the text with comments showing the remaining work in this respect.

2) Concerning the dynamic spiking model, for the t_r and t_f times, it would be good for authors to compare their model with one recently published and not referenced despite many similarities:

T. Rosca, F. Qaderi and A. M. Ionescu, "High Tuning Range Spiking 1R-1T VO₂ Voltage-Controlled Oscillator for Integrated RF and Optical Sensing," ESSCIRC 2021 - IEEE 47th European Solid State Circuits Conference (ESSCIRC), 2021, pp. 183-186, doi: 10.1109/ESSCIRC53450.2021.9567761.

Please include a discussion in the paper what are the differences (any novelty or not?) and similarities of the spiking models.

3) It is good that an SNR is now defined and some benchmarks included (Table). However, the claimed high sensitivity of 151.74 kHz/N, 0.13kHz/Lux and 2.8 kHz/°C in tactile, optical, and temperature perception, respectively are not compared directly with existing dedicated state of the art integrated sensors. Can the author comment if such sensitivities (and selectivities as well) are comparable, better or poorer than the ones of existing sensors (at least as order of magnitude)? Authors should clarify better the advantages of their sensors (based on the choices made).

In conclusion, the paper can be published only after addressing the few last modifications suggested.

Reviewer #2 (Remarks to the Author):

The authors have addressed my comments and improved the text and discussion. The manuscript can now be accepted for publication. The results are nice and will surely attract the attention of the readership of Nature Communications.

Reviewer #3 (Remarks to the Author):

The authors have revised the manuscript properly and I do not have any other concern. In my opinion, this manuscript is suitable for publication in Nature Communications.

MS No: NCOMMS-22-05908A

Title: A Calibratable Sensory Neuron Based on Epitaxial VO₂ for Spike-based Neuromorphic Multisensory System

Response to the editor and the reviewers

We would like to sincerely thank the editor for the kind consideration of our manuscript and are pleased to find that reviewers #2 and #3 have recommended publication. In this newly revised submission, we have carefully revised the manuscript, in light of the remaining suggestions from reviewer #1. The point-to-point responses and changes made are listed below.

Comments from Reviewer #1

Overall Remarks: The authors have addressed in details many of the important raised questions and the paper is improved, this is very appreciated.

Our response: We would like to sincerely thank the reviewer for the very detailed and constructive suggestions. Our point-to-point responses to the comments and the corresponding changes are shown as follows.

1) I appreciate that the power limit is discussed now better in the paper. On the other hand, it appears that the power per spike is not yet very competitive for neuromorphic applications and even matching the state of the art (few nanoJoule reported is far from neuromorphic literature showing few pJ/spike as state of the art) However, the discussion provided and the rationale how to scale are now

improved. Please try to carefully reflect this gap in the text with comments showing the remaining work in this respect.

Our response: We would like to greatly thank the reviewer for the detailed comment. To address this question, we have added the following discussion in page 11 of the revised manuscript: *“This value is still lower than the state of the art reporting few pJ/spike (Ref. 28)”*, and *“Future work will focus on continued optimization of the growth conditions for VO₂ films and scaling the size of the devices to further reduce the energy consumption”*.

2) Concerning the dynamic spiking model, for the tr and tf times, it would be good for authors to compare their model with one recently published and not referenced despite many similarities:

T. Rosca, F. Qaderi and A. M. Ionescu, "High Tuning Range Spiking 1R-1T VO₂ Voltage-Controlled Oscillator for Integrated RF and Optical Sensing," ESSCIRC 2021 - IEEE 47th European Solid State Circuits Conference (ESSCIRC), 2021, pp. 183-186, doi: 10.1109/ESSCIRC53450.2021.9567761.

Please include a discussion in the paper what are the differences (any novelty or not?) and similarities of the spiking models.

Our response: We would like to greatly thank the reviewer for pointing out this important reference, and sorry for not citing it in the previous version of manuscript, because we really didn't find this reference. Now it has been included in the references as Ref. 56.

Our model indeed has similarities with this prior work - both circuits are essentially relaxation oscillators that rely on a volatile resistive switching VO₂ device to enable self-sustained oscillations. The capacitor in the circuit undergoes a charging phase and a discharging phase within a period, and the voltage across the VO₂ device fluctuates between V_{th} and V_{hold} . Therefore, both circuit models share a similar mathematical form.

However, our model still has many differences. For example, our circuit consists of a load resistor, a capacitor, and a VO₂ device, in which the capacitor charges through the load resistor and discharges through the VO₂ device. Both charging and discharging currents depend on the time-varying voltage across the capacitor. In contrast, the circuit in Ref. 56 features a MOSFET, other than the capacitor and the VO₂ device. The capacitor charges through the VO₂ device and discharges through the MOSFET instead. While the charging current varies with the voltage across the capacitor, the discharging current is constant, as the n-MOSFET with a fixed gate voltage is simply a constant current source. Hence, the two circuit models have different RC time constants, namely, $(R_{VO_2} || R_L)C_m$ in our model vs. $R_{VO_2}C_m$ in Ref. 56. Besides, they also have different voltage dividing terms across the capacitor, namely, $\frac{R_{VO_2}}{R_L + R_{VO_2}}V_{in}$ in our model vs. $R_{VO_2}I_{D,MOS}$ in Ref. 56. These differences are reflected in the t_f and t_r expressions.

Although our model (together with those in Ref. 54-55) and the one in Ref. 56 are similar in general, the main novelty of our model lies in the fact that we have taken into account the scaling resistance and the calibration resistance (Eq. 8 of the main text). This is in line with the novelty of our work, in which we demonstrated an artificial

neuron that is capable of adapting to different kinds of sensors with varied resistance levels.

To address this question, the prior work is included as Ref. 56 and we have added the following discussion in page 20: *“This model is similar to the one given in Ref. 54-56, which are essentially relaxation oscillators relying on volatile threshold switching devices to enable self-sustained oscillations. However, it is worthwhile noting that the detailed time constant and voltage dividing terms across the capacitor are different due to their different circuit designs, and more importantly, this model takes into account the scaling resistance and the calibration resistance.”*

56. Rosca, T., Qaderi, F. & Ionescu A. M. High Tuning Range Spiking 1R-1T VO₂ Voltage-Controlled Oscillator for Integrated RF and Optical Sensing. In: *ESSCIRC 2021 - IEEE 47th European Solid State Circuits Conference (ESSCIRC)* (2021).

3) It is good that an SNR is now defined and some benchmarks included (Table). However, the claimed high sensitivity of 151.74 kHz/N, 0.13kHz/Lux and 2.8 kHz/°C in tactile, optical, and temperature perception, respectively are not compared directly with existing dedicated state of the art integrated sensors. Can the author comment if such sensitivities (and selectivities as well) are comparable, better or poorer than the ones of existing sensors (at least as order of magnitude)? Authors should clarify better the advantages of their sensors (based on the choices made).

Our response: We would like to thank the reviewer for the detailed comment. While achieving impedance matching with different sensors, the epitaxial VO₂ based sensory neurons in this work still achieve high sensitivity in different sensing modes, namely, 151.74 kHz/N, 0.13 kHz/Lux, and 2.8 kHz/°C for tactile, optical, and temperature perception, respectively. Given that our pressure sensor is 2 cm in diameter, the calculated sensitivity to pressure is 47.67 kHz/kPa, which is slightly lower than the 60.8 kHz/kPa reported in Ref. 30. This can be improved by increasing the sensitivity of the pressure sensor itself. The important point is that our spiking sensory neuron can be matched with different kinds of sensors, which is a significant advantage over existing studies. As for the sensitivity to temperature and light intensity, there seem to be no prior works reporting such metrics that can serve as the background for direct comparison, to the best of our knowledge.

We want to further note that our approach has advantages in the following aspects:

1) Compared with existing works in the literature, our study achieves multiple perception modalities including pressure, light, temperature and curvature for the first time, which is due to the successful solution of the impedance matching problem between sensors and neurons using our neuron circuit. Neurons can fire only when the resistance of the sensor is within a specific range. However, the resistance ranges between different sensors are very different, which is the reason why it is difficult for previous works to achieve multiple sensing modes. To match impedance with various sensors, we added a scaling resistor and calibration resistor in our neuron circuit to allow the neuron to adapt to different sensors, which is a significant advantage of our

approach and is not seen in other studies.

2) We have defined and analyzed the signal-to-noise ratio (SNR) of the spiking sensory neuron, which has not been reported in other state-of-the-art spike-based sensory neurons. The high crystalline quality of epitaxial VO₂ has addressed the fundamental cycle-to-cycle and device-to-device variation issues in sensory neurons, and the resultant excellent uniformity of our devices gives rise to excellent SNRs of 33.66 dB, 31.90 dB and 29.92 dB in tactile, optical and temperature sensing, respectively.

3) While achieving impedance matching with various sensors, our spiking sensory neuron can still achieve high sensitivity in tactile, optical, and temperature perception. To the best of our knowledge, our study is the first to report the sensitivity of spiking neurons to illuminance and temperature.

To address this question, we have added the following discussion in page 15:
“Given that our pressure sensor is 2 cm in diameter, the calculated sensitivity to pressure is 47.67 kHz/kPa, which is slightly lower than the 60.8 kHz/kPa reported in Ref. 30. This can be improved by increasing the sensitivity of the pressure sensor itself. As for the sensitivity to temperature and light intensity, there seem to be no prior works reporting such metrics that can serve as the background for direct comparison, to the best of our knowledge. The important point is that our spiking sensory neuron can be matched with different kinds of sensors, which is a significant advantage over existing studies.”

30. Fang, S. L. *et al.* An Artificial Spiking Afferent Neuron System Achieved by 1M1S for Neuromorphic Computing. *IEEE Trans. Electron. Devices* 1-7 (2022).